# Automated detection and analysis of surface calving waves with a terrestrial radar interferometer at the front of Eqip Sermia, Greenland

Adrien Wehrlé[1], Martin P Lüthi[1], Andrea Walter[1], Guillaume Jouvet[1], and Andreas Vieli[1]

[1]Institute of Geography, University of Zurich, Zurich, Switzerland

**Correspondence:** Adrien Wehrlé (adrien.wehrle@geo.uzh.ch)

**Abstract.**

Glacier calving is a key dynamical process of the Greenland Ice Sheet and a major driver of its increasing mass loss. Calving waves, generated by the sudden detachment of ice from the glacier terminus, can reach tens of meters of height and provide very valuable insights to quantify calving activity. In this study, we present a new method for the detection of source location, timing and magnitude of calving waves using a terrestrial radar interferometer. This method was applied to 11500 one-minute interval acquisitions from Eqip Sermia, West Greenland, in July 2018. During seven days, more than 2000 calving waves were detected, including waves generated by submarine calving which are difficult to observe with other methods. Quantitative assessment with a Wave Power Index (WPI) yields a higher wave activity (+49 %) and higher temporally cumulated WPI (+34 %) in deep water than under shallow conditions. Subglacial meltwater plumes, occurring 2.3 times more often in the deep sector, increase WPI and the number of waves by a factor 1.8 and 1.3 respectively in the deep and shallow sector. We therefore explain the higher calving activity in the deep sector by a combination of more frequent meltwater plumes and more efficient calving enhancement linked with better connections to warm deep ocean water.

## 1 Introduction

Many outlet glaciers of the Greenland Ice Sheet have undergone rapid retreat, thinning and flow acceleration within the past two decades (e.g., Moon et al., 2012; Enderlin et al., 2014; King et al., 2020) and have become important contributors to the observed increasing mass loss rates of the Greenland Ice Sheet (Shepherd et al., 2012; IPCC, 2013; Shepherd et al., 2020). Despite many sophisticated studies, major limitations remain in the understanding of the dynamics of ocean-terminating glaciers (Vieli and Nick, 2011; Straneo and Heimbach, 2013; Catania et al., 2020), characterized by a very high temporal and spatial variability (Moon and Joughin, 2008). While field measurements provide the high resolution required for capturing the associated processes (e.g., Walter et al., 2020), their limited coverage during short measurement campaigns reduces them to snapshots of the long-term behavior of tidewater glaciers. On the other hand, repeat space-borne observations offer long time series, but at a limited spatial and temporal resolution.

Glacier calving, a sudden fracture phenomenon that releases large quantities of ice to the proglacial fjord during short-lived events, has been identified as an important factor in the dynamics of tidewater glaciers (e.g., Joughin et al., 2004; Luckman

et al., 2006; Nettles et al., 2008; Amundson et al., 2008). This process has been studied using various methods including seismic source inversion (Walter et al., 2012; Sergeant et al., 2019), detailed numerical modeling (e.g., Benn et al., 2017; Mercenier et al., 2020), underwater acoustics (Glowacki and Deane, 2020), and radar interferometry (e.g., Lüthi and Vieli, 2016; Xie et al., 2019; Cassotto et al., 2019; Walter et al., 2020; Cook et al., 2020; Kane et al., 2020). Often, these methods are complemented with high-rate time-lapse photography (e.g. Dietrich et al., 2007; Amundson et al., 2008; Lüthi et al., 2009; Minowa et al., 2018).

Glacier calving events generate ocean waves by falling ice chunks, rotational detachment of full-thickness ice blocks, or buoyant up-rise of submerged ice. Such calving waves can reach heights exceeding $50\,\mathrm{m}$ and create damaging tsunami waves upon run-up on the shores (Reeh, 1985; Lüthi et al., 2009; Amundson et al., 2010; Lüthi and Vieli, 2016). While the timing of calving waves is accurately captured by tide gauges or moorings (e.g., Amundson et al., 2010; Minowa et al., 2018), reconstructing the source location from wave measurements alone is difficult due to irregular spreading patterns associated with complicated fjord bathymetry. A way to overcome this limitation is the use of time lapse photography, which, however, requires manual identification (Minowa et al., 2018).

Studying the origin, mechanism, source impact and spreading of surface calving waves in space and time remains a challenge due to their transient characteristics, a variety of source mechanism, and the heterogeneous and dynamic propagation environment in iceberg-covered fjords. Here, we present a method to investigate calving event source positions by back-tracking wave trains captured with a terrestrial radar interferometer (TRI). To our knowledge, this is the first application of a TRI to observe and quantify surface calving waves. The method was successfully applied to the detection of more than 2000 calving events within a data set acquired over 7 days at Eqip Sermia, West Greenland. In the following, we first present the study site and data before describing the different steps of the method. We proceed to analyze the associated results and discuss the challenges linked to calving wave detection and method improvements. Finally, we extend our analysis by combining our results of calving wave activity with a detection of visible melt water plume occurrences.

## 2  Study site

Eqip Sermia (69.80°N / 50.22°W; Fig. 1) is a medium size outlet glacier situated on the West coast of the Greenland Ice Sheet. After a century of slowly varying terminus positions, a rapid retreat and flow acceleration started in the year 2000 (Lüthi et al., 2016), associated with high calving activity (Walter et al., 2020). The calving front is about $3.5\,\mathrm{km}$ wide with a height above the water line between 50 and $170\,\mathrm{m}$. The whole front is grounded and can be divided into a shallow central sector and a deeper southern sector (approximate water depths of 0-20 m and 70-100 m, respectively) by extrapolation of bathymetric surveys (Rignot et al., 2015; Lüthi et al., 2016), ice cliff geometry, and calving activity (Walter et al., 2020). Due to its rapid flow dynamics Eqip Sermia is extremely crevassed and mostly inaccessible for in-situ measurements.

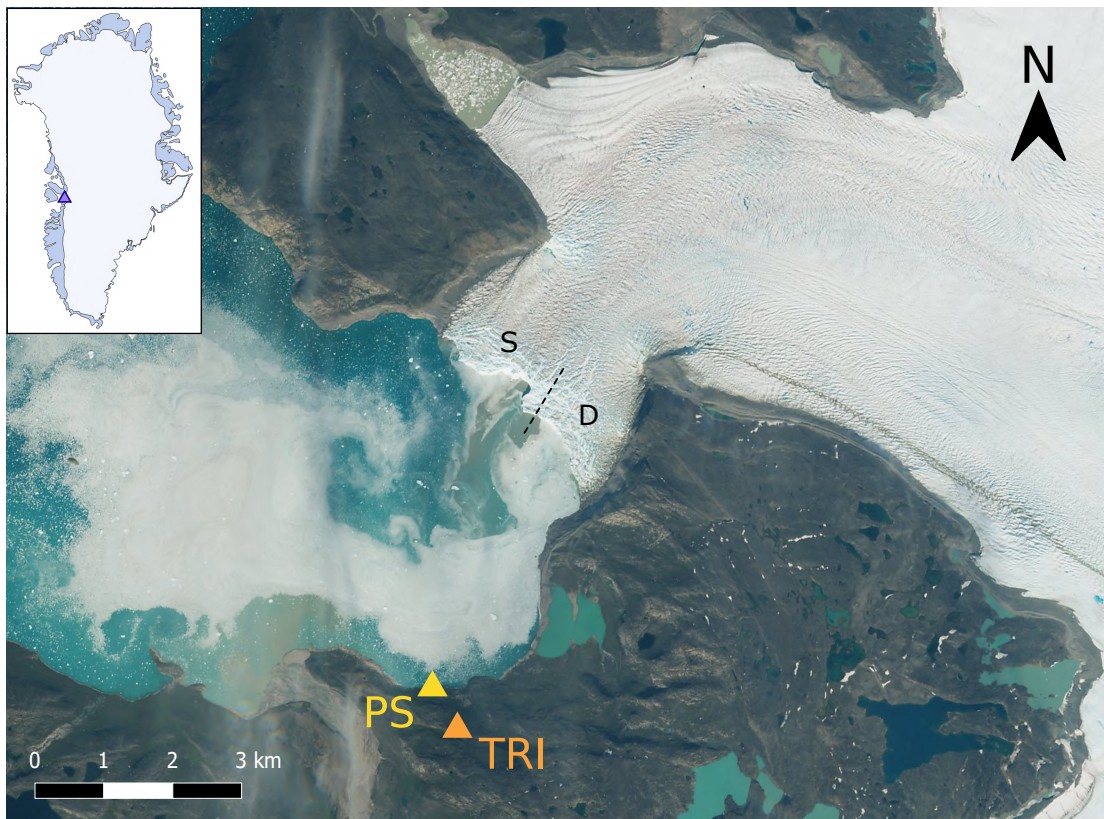

**Figure 1.** Eqip Sermia in West Greenland flows into a shallow fjord. Clearly visible are freshly calved icebergs and drifting ice debris pushed out by the brownish melt water plume (under the letter S). The positions of the terrestrial radar interferometer (TRI) and the pressure sensor (PS) are indicated by orange and yellow triangles. Sectors ending in shallow (S) and deep (D) waters are separated by a black dashed line at the calving front. Background: Sentinel-2A scene from 19 July 2018 (Copernicus Sentinel data 2018, processed by ESA).

## 3  Methods

### 3.1  Instrumentation

A terrestrial radar interferometer (TRI; Werner et al., 2008; Caduff et al., 2014) was used during field campaigns in the summers 2014-2019 to monitor the calving activity of Eqip Sermia. The installation site on bedrock is $150\,\mathrm{m}$ above sea level at a distance of $4.5\,\mathrm{km}$ from the calving front ($69.7523°\mathrm{N}\,/\,50.2520°\mathrm{W}$; Fig. 1). The TRI system was a Gamma Portable Radar Interferometer (GPRI) operating at a wavelength of $\lambda = 17.4\,\mathrm{mm}$ (Ku-Band, 17.2 GHz). It is composed of one transmitter and two receiver antennas, rotating along the vertical on a precision astronomical mount. The range resolution is approximately $0.75\,\mathrm{m}$ while the azimuth resolution is 0.1 degrees, corresponding to $7.9\,\mathrm{m}$ at a slant range of 4.5 km (Werner et al., 2008). In this study radar intensity measurements from the upper antenna, acquired in July 2018, are used. Measurements from the lower antenna are almost identical, although show a slightly fainter signal contrast.

A pressure sensor was installed on the shore in front of the TRI to record the water pressure in the fjord (3.5 km away from the calving front; position PS in Fig. 1). Water pressure was recorded every 4 seconds and was converted to water surface height to retrieve the amplitude and timing of surface calving waves generated at the glacier front.

## 3.2   Data analysis

### 3.2.1   Calving wave detection

The TRI registered in a one-minute interval the signal strength and phase of reflecting natural surfaces on the glacier and the fjord, such as ice faces, rocks, and icebergs. The raw radar acquisitions were stored as complex numbers in $598 \times 11184$ pixel arrays. The method described in the following is focused on the analysis of the signal strength.

Figure 2a shows an example of a captured calving wave originating from the central glacier front, displayed as signal intensity. The ensuing concentric wave train is well visible as bright signal back-scatter from many small ice fragments floating 75 in the fjord. Several large icebergs cast shadows from the radar illumination from the left.

We developed a novel algorithm to automatically detect calving waves, their origin and a measure of their magnitude in time series of radar acquisitions named TeRACWA (Terrestrial Radar Assessment of Calving Wave Activity). The algorithm was implemented in the Python programming language, using the scipy and multiprocessing libraries for signal processing and parallel algorithm execution (Virtanen et al., 2020). Figure 3 shows the sequence of processing steps, the details of which are 80 described below. TeRACWA works on the signal intensity from the raw data acquisitions in radar geometry (range and azimuth angle).

– **Step 0: Extraction of the glacier terminus.** Based on the mean signal intensity from the TRI over the field season, an outline of the average calving front position was manually defined.

– **Step 1: Masking.** A pixel mask was applied to restrict processing to a region of interest (ROI) and to limit the influence 85 of external features such as land and glacier areas. The upstream (right in Fig. 2a) delineation of the ROI is the outline of the glacier front extended 200 m (approx. 270 pixels) up onto the glacier, including the area of glacier front variations during the field season. The downstream (left in Fig. 2a) delineation is the projection of the glacier front into the fjord by 3000 pixels (approx. 2.2 km). With this definition, the ROI has the same number of pixels for every azimuth line (horizontal image rows) and covers most of the fjord area in front of the glacier.

– **Step 2: Background signal reduction.** Two consecutive intensity images were subtracted for change detection in the fjord and to simultaneously cancel out the signal from stable ice mélange and icebergs.

The following processing steps 3-6 were applied to each resulting differenced image. More specifically, step 3 was applied to the signal strength of each individual azimuth line of each differenced image, further referred to as one-dimensional array.

– **Step 3: Spectral analysis.** The power spectrum of each one-dimensional array was obtained with a Fourier transform. A 95 band-pass filter with empirically selected low and high cutoff wavelengths was then applied in order to restrict the search

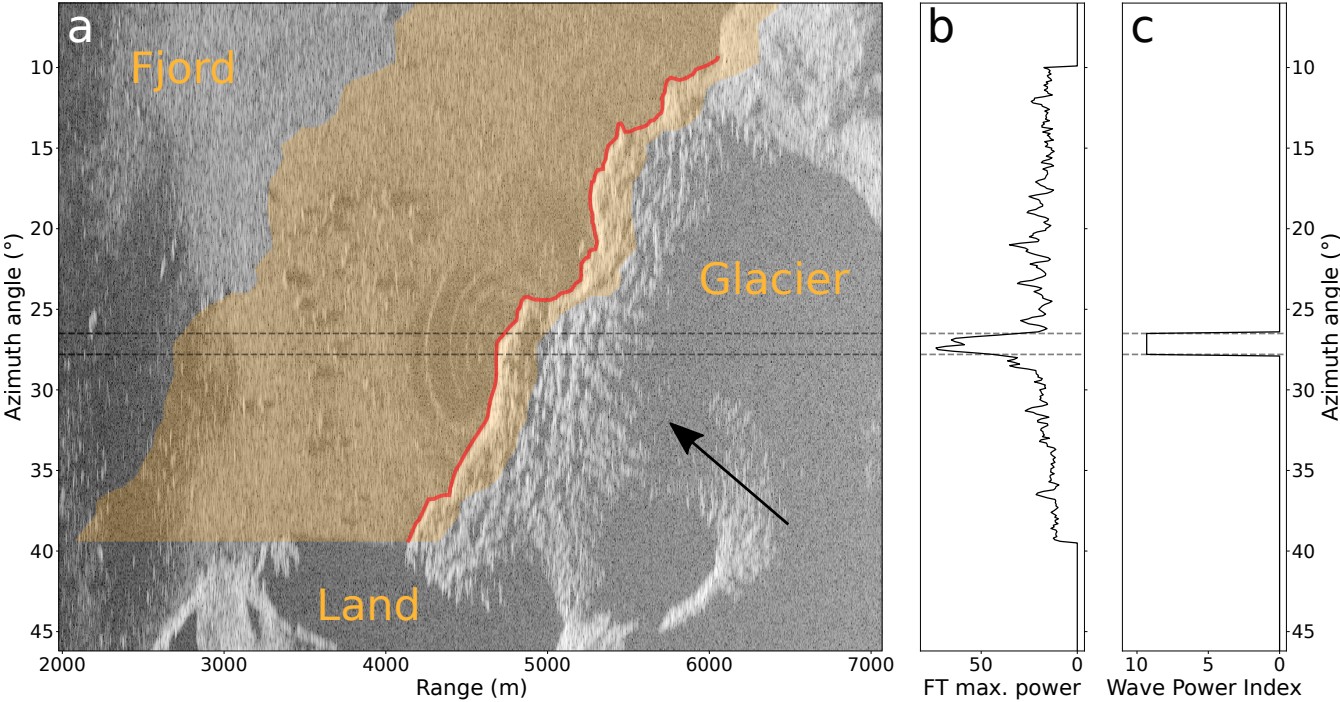

**Figure 2.** Example of calving wave detection. (a) Radar image of July 7, 2018 at 06:21:00 UTC represented as the logarithm of the signal strength, visualised using the matplotlib Python library (Hunter, 2007). The black arrow indicates the ice flow direction, and the red line the outline of the glacier front. The translucent orange area shows the region of interest (ROI). (b) Maximum power of the Fourier transform by azimuth line obtained after the spectral analysis (step 3). (c) Wave power index (WPI) computed by TeRACWA (step 6). The two dashed lines indicate the boundaries of the wave detection on panels (a) and (b). The part of the calving front in between the lines corresponds to the wave source area.

of frequencies that correspond to calving waves. This filtering reduced atmospheric noise and ocean surface roughness (high frequencies) as well as wave reflections on the shores and iceberg rotation or breakup (low frequencies).

The low-cutoff wavelength was selected by analyzing the power spectrum of 60 different signals, each of which being the average of 4 neighboring azimuth lines. Half of these signals were taken from acquisitions containing calving waves with azimuth lines around the source locations (Fig. 4b), the other half was composed of randomly selected background signals (Fig. 4a). With support from high-rate timelapse images acquired at the calving front (Walter et al., 2021), we determined that 5 out of 17 wave signals visible on images were associated with submarine calving events (the other 13 waves occurred while the camera was not running). Comparing the average of the two types of power spectra, a significant increase in power appears at a wavelength exceeding 13.7 m for spectra including calving waves (Fig. 4c, d). To avoid a too restrictive threshold the low-cutoff wavelength was set to 12.3 m (90 % of the initial value). As a high-cutoff, a wavelength of 800 m was selected since it suppresses reflections from the shores. For each one-dimensional

array, the maximum power within the restricted frequency range was computed. The resulting two-dimensional matrix of maximum power was assembled with azimuth lines as lines and time steps as columns.

– **Step 4: Normalization.** The resulting time series of maximum power per azimuth line were normalized individually by subtracting the mean and dividing by the standard deviation. This normalization corrects for the heterogeneous addition of signal power from the glacier front and upstream crevasses. The net effect of this step is an even signal power distribution among the azimuth lines.

– **Step 5: Wave detection.** Waves are characterized by a quick temporal power increase, followed by a dissipation through several azimuth lines. To detect position and timing of waves, a 2 dimensional peak detector based on the maximum filter function of the Scipy multi-dimensional image processing package (ndimage) was used. To discern the waves from background signal, the heterogeneous and rapidly changing properties of the ocean surface had to be taken into account. To this end, the minimum value within a duration of $\pm 5$ minutes (i.e. 10 acquisitions) around a given wave was determined and considered as the background signal level. The wave power index (WPI) was then defined as the peak prominence, i.e. the peak height above the immediate surrounding background signal. The location and width of the wave in the azimuth dimension (along front) was given as the span over the peak full width at half prominence (Fig. 2c). The location therefore consisted of the azimuth lines over which a given wave was detected without further information about the range span. Since the peak detection also selects minor peaks associated with noise, a threshold had to be set on the minimum prominence associated with a wave. To this purpose the Kneedle algorithm (Satopaa et al., 2011) was used. Considering the number of waves as a function of the WPI threshold, it was possible to detect a fast increase in the number of detected waves when decreasing the threshold value in the form of a curve inflexion. The threshold associated with this change in data behaviour identified at a WPI of 4.5 corresponds to the lowest threshold applicable before a significant increase in the number of waves associated with false detection of noise, the latter exhibiting an average WPI of 1.9 in the data set here analysed.

– **Step 6: Correction.** To prevent the detection of several power peaks for the same wave on one acquisition that are potentially linked with heterogeneous bay properties, a correction was applied. An illustration of this process is given in Figure 2b. In this example, two peaks are associated with the detected wave, but only the maximum peak at three quarters of the prominence was selected. As the peak detection was applied in two dimensions (azimuth and time), this correction also prevented the detection of several peaks for a same wave through time. This pattern was identified for waves associated with low signal-to-noise ratios but remained rarely observed.

The final product consists of a catalog of wave generation times and along-front locations, as well as associated wave magnitudes quantified by an empirical wave power index (WPI) throughout the field season. From this catalog, spatially and temporally cumulated, and averaged WPI values were computed. The spatially cumulated WPI consisted of the sum of WPI values along the calving front at a given timestep, further applied to all timesteps. The temporally cumulated WPI consisted of the sum of WPI values through time for a given azimuth line (image horizontal row), further applied to all azimuth lines.

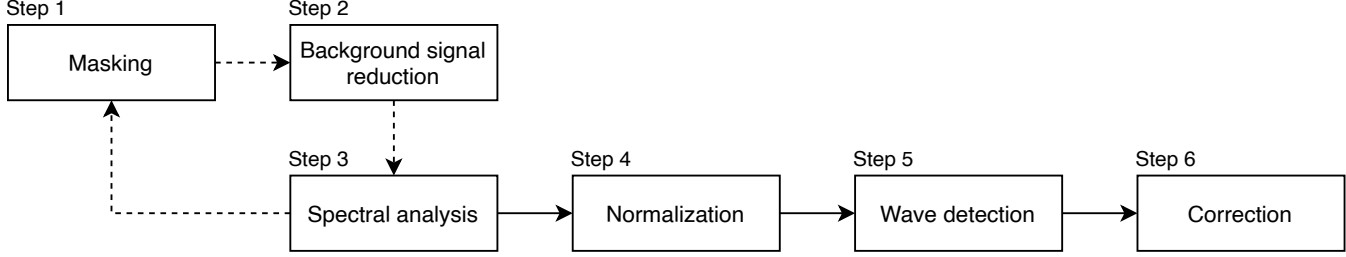

**Figure 3.** Steps of the wave detection algorithm TeRACWA (Terrestrial Radar Assessment of Calving Wave Activity). Steps linked by dashed and solid arrows are respectively applied to each TRI acquisition and to the resulting data set.

**Figure 4.** Determination of the frequency high-cutoff by analysing the power spectra of (a) 30 random background signals and (b) 30 wave signals. (c) Power spectral density (PSD) averages for the selected background (gray) and wave (blue) signals. (d) Difference between the two curves shown in (c). The gray dotted lines indicate the low and high cutoff wavelengths (13.7 m and 800 m, respectively).

### 3.2.2 Wave energy quantification

The water pressure sensor data shows that the fjord was very calm without the forcing from calving events. Background waves driven by winds or ocean swells were mostly absent. For each calving event the water height data shows direct and reflected waves of many amplitudes. To obtain a quantitative relation between the WPI detected by TeRACWA and wave amplitudes measured with the pressure sensor, we calculated a quantity called "Integrated Wave Height Squared" (IWHS) as a measure of wave energy reaching the shore. Simpler measures, such as maximum wave height, proved to be less suitable as they do not capture the temporal evolution of the wave signal. The IWHS was computed during an interval of $\Delta t = \pm 3\,\mathrm{min}$ around the generation time $t_w$ of the wave of maximum height. Water level readings $H(t)$ were therefore summed such that

$$\mathrm{IWHS} = \int\limits_{t_w - \Delta t}^{t_w + \Delta t} |H(t)|^2 \, dt \,, \tag{1}$$

which has units of $\mathrm{m}^2\mathrm{s}$.

The integration interval $\Delta t$ was set as a compromise to capture most of the wave energy while keeping a limited overlap with following waves. Attempts to determine an integration interval specific to the wave time span were carried out with the aim of preventing any contribution from surrounding waves but the method efficiency remained too low for the latter to be integrated in the analysis. The maximum wave height $H_{max}$ at the pressure sensor of every wave recorded during the TRI operation was first determined. The 50 highest waves of the resulting data set were manually assigned to the corresponding WPI values calculated by the TeRACWA algorithm. It was not possible to achieve such an association with confidence for smaller waves.

### 3.2.3 Meltwater plume detection

With the aim of studying the relation between meltwater plume occurrence and calving wave activity we manually quantified visible meltwater plumes. Plume footprints are clearly discernible on optical imagery as growing sediment-rich hence brownish areas close to the water surface, and whether the fjord is ice covered or not (Figs. 1 and 5a). However, the detection of such evolving patterns from satellite optical imagery can only be carried out with a limited temporal resolution of several days. On high-resolution TRI intensity images, plumes surrounded by ice mélange are clearly visible as they push the dense ice debris coverage (almost permanent during the 2018 field campaign) from the calving front into the fjord creating open water areas. An example of plume evolving in the deep sector as seen on consecutive TRI intensity images is shown in Movie SC1. To suppress very high frequency variations and to a lower extent, limit manual detection, we first produced 195 hourly averaged TRI signal intensity plots. On the hourly averaged images, as well as on consecutive images, the distinctive patterns linked to the evolution of such meltwater plumes made their detection unambiguous and prevented the false detection of wind-driven open water areas and open water areas following calving events. For each image of this catalog of hourly averaged intensity images we then manually determined the azimuth lines at which melt water plumes were visible at the calving front. Only the presence or the absence of visible melt water plumes was retrieved, other characteristics, such as the footprint area, were not

monitored. Based on the resulting data set the time fractions of occurrence of visible meltwater plumes during the observation period were determined for each pixel along the calving front.

## 4 Results

### 4.1 Analysis of calving wave activity

The TeRACWA algorithm was applied to co-registered TRI data from the 2018 field season from July 7 to 15. Within the 11479
acquisitions in one minute-interval during 7.49 days, a total of 2418 calving waves were automatically detected, resulting in an average of 13.4 calving waves per hour. Figure 5b summarizes the calving wave detection results during the field season with colored bars indicating the timing and source location along the front. The colors indicate the WPI as a measure of wave magnitude. Surrounding panels show temporal (Fig. 5c) and spatial (Fig. 5d) variability of calving waves from the different source locations (Fig. 5a).

Both the spatial and temporal variability of calving wave activity are large, with episodic quiet and active phases along different parts of the calving front. Due to the distinctly different characteristics in ice cliff geometry and water depth, the calving front can be divided into sectors with shallow and deep water which exhibit different calving behaviour and event size statistics (Fig. 1; Walter et al., 2020).

The results in Figure 5d clearly show the difference between the deep and the shallow sectors in terms of number of waves
and temporally cumulated WPI. More calving waves were detected within the deep sector (+10 %), with a +49 % higher calving frequency when normalized by sector width. The difference is also apparent in the wave magnitude: both the temporally averaged WPI (+4 %) and the temporally cumulated WPI (+34 %) are larger in the deep sector. The relative difference in temporally cumulated WPI between the two sectors is more than 8 times larger than for the temporally averaged WPI which is explained by a higher number of events associated to a higher average WPI. Indeed, both variables are contributing to an
increased difference in temporally cumulated WPI between the two sectors while the temporally averaged WPI is normalized by the number of events. However, waves detected in the deep sector are more localized, as they show a smaller width (-9 %) which can be linked to the contrasting water depths of the two sectors. The associated numbers are summarized in Figure 6. The differences between the two sectors for the variables presented in this figure have been determined to be statistically significant using a t-test or the null hypothesis that two independent samples (here the data sets of the two sectors) have identical average
(expected) values. The different t-tests yielded p-values below 0.001 and t-statistics from 3.4 to 21.1.

Figure 5b shows that the calving wave activity is highly variable in time. The spatially cumulated WPI (Fig. 5c), consisting in sums of WPI values along the front within consecutive 20-minute intervals, exhibits strong and rapid variations. During the measurement period of seven days the spatially cumulated WPI, resampled to twelve hours, shows a positive trend (time correlation of 0.68) with more frequent and more localized calving events. The largest calving events occur during low activity
periods. However, no relation has been found between the number of waves or spatially cumulated WPI within a time windows from 5 minutes to 1 hour around each detected wave and their respective WPI. At a daily scale, spatially cumulated WPIs and number of waves per hour recorded from 4 am to 10 pm are on average 9 % higher than during the rest of the day. However,

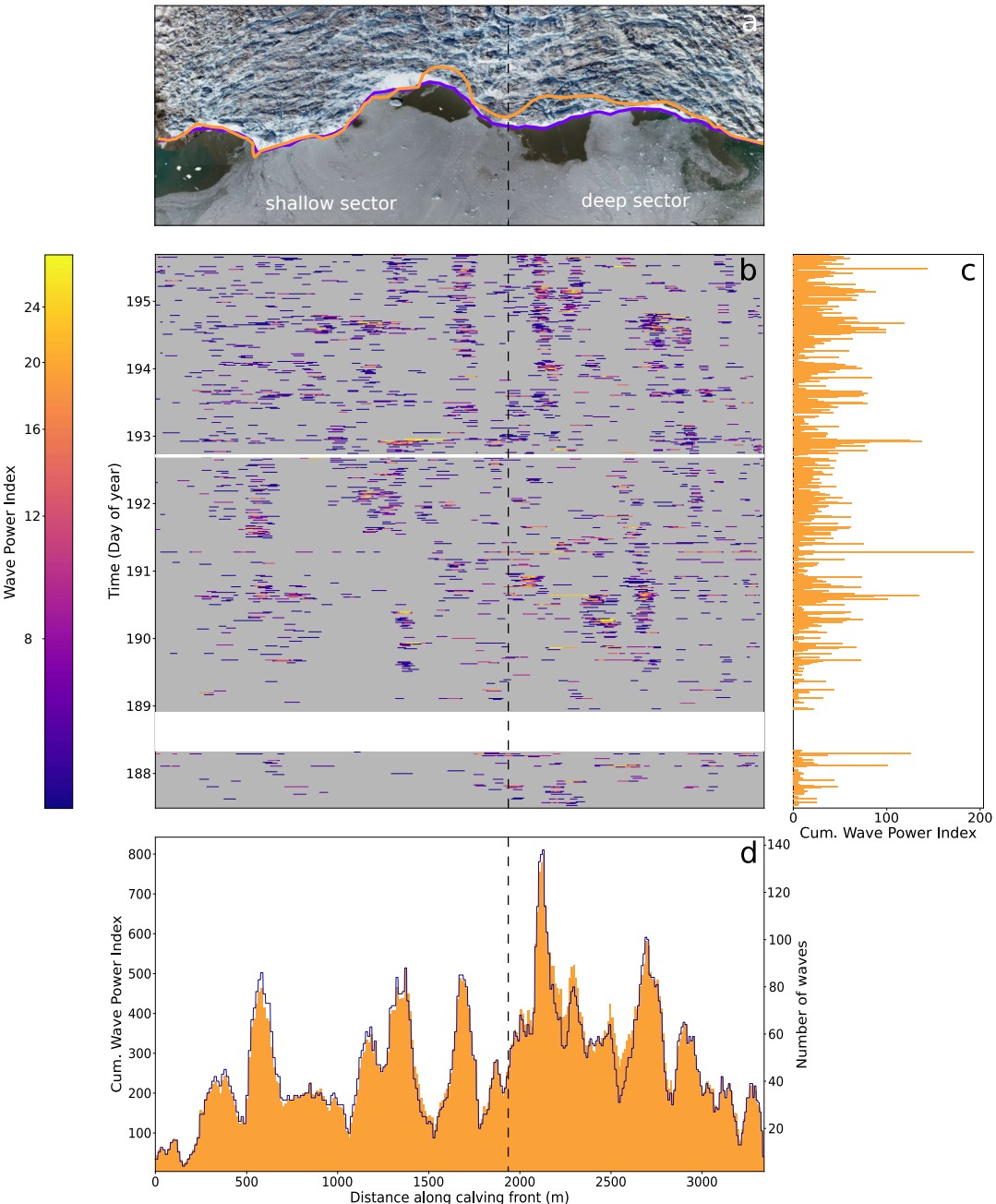

**Figure 5.** Results of the automatic wave detection by TeRACWA during the 2018 field season. (a) Orthomosaic of the calving front acquired on July 6th using an unmanned aerial vehicle (Jouvet et al., 2019) combined with the outline of the calving front at the beginning (July 6th; purple curve) and end (July 14th; orange curve) of the TRI observation period (Walter et al., 2021). The vertical dashed line indicates the transition from the shallow sector to the deep sector. (b) Wave power index (WPI) of the waves detected over time (20 minute stacks) along the calving front. White areas correspond to data gaps. (c) Spatially cumulated WPI over time with WPI values from (b) summed along the front. (d) Temporally cumulated WPI (orange bars) and number of waves (purple line) along the calving front, over the entire period.

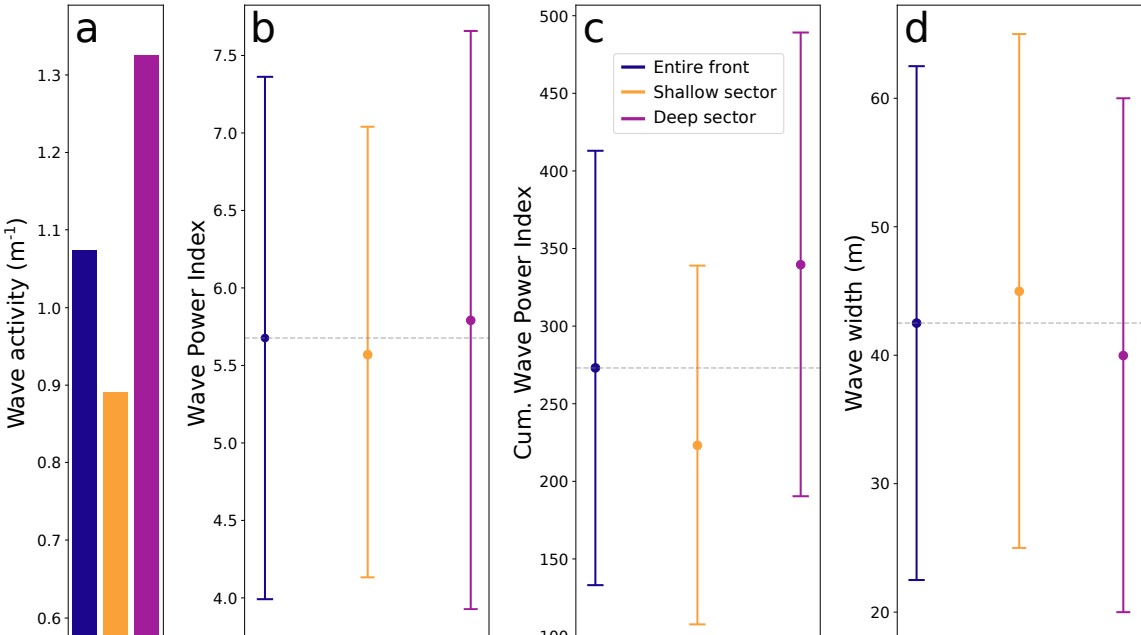

**Figure 6.** (a) Wave activity, (b) temporally averaged WPI, (c) temporally cumulated WPI along the calving front and (d) wave width at generation point. The wave activity is obtained by normalizing the number of detected waves by the sector width. For panels b, c and d, sector-averaged values are shown as dots and the associated standard deviations for each sector as bars ($\pm 1\sigma$). Gray horizontal lines indicate mean values along the entire front.

no relation has been found between WPI or wave activity and tide or air temperature (correlation coefficients in between -0.08 and 0.14). Air temperature and relative humidity measurements as well as tide heights derived from the pressure sensor data are presented in Figure SA1 and associated to the spatially cumulated WPI throughout the acquisition period.

To further quantify statistical characteristics of the observed wave activity, recurrence times $t_r$ for waves of the same or higher WPI were determined (discretization with bins of width 0.5 WPI). Figure 7 shows wave recurrence times from 4.8 minutes for a WPI equal or above 4.5 to 55.4 hours for a WPI equal or above 23. These data points were fitted with a power law using non-linear least squares

$$t_r = t_0 \, \mathrm{WPI}^\alpha,\tag{2}$$

where $t_0 = 10^{-5}\,\mathrm{h}$ is a scaling factor and the resulting power is $\alpha = 3.1$.

### 4.2 Relation to water surface height

Figure 8 shows the WPI versus maximum wave height and IWHS, respectively, for the 50 highest waves detected in the pressure sensor data set. The biggest wave ($2.64\,\mathrm{m}$) with highest IWHS ($69.7\,\mathrm{m^2s}$) corresponds to the highest WPI determined over the period (28.8). On the other hand, a WPI of 6.2 corresponds to a wave height of $0.45\,\mathrm{m}$ and a IWHS of $3.3\,\mathrm{m^2s}$.

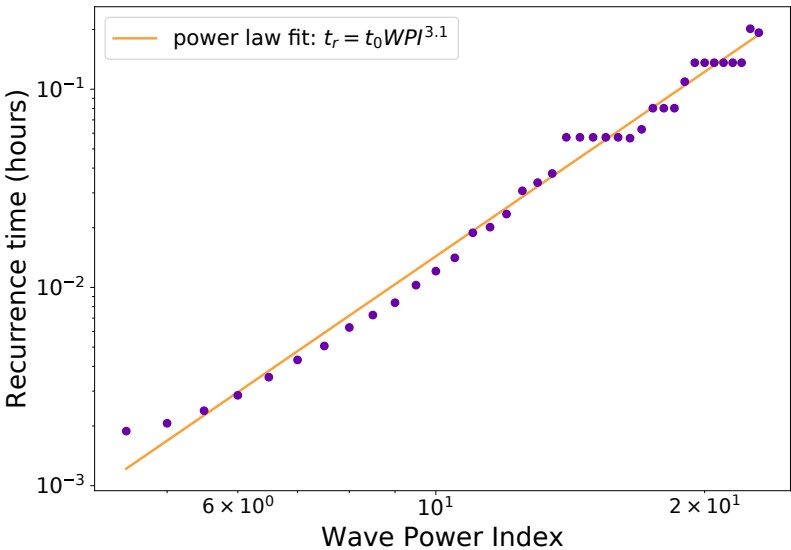

**Figure 7.** Wave recurrence time for a range of WPI values. The orange line corresponds to a power law fit using non-linear least squares where $t_0$ is a scaling factor ($10^{-5}$).

Examples of associations between waves detected by TeRACWA and wave heights derived from the pressure sensor data are presented in Figure SB1 for the three highest WPI determined during the acquisition period. A linear least-squares fit on the data (Fig. 8a) gives correlation coefficients of $R = 0.81$ for all considered waves and $R = 0.87$ for waves originating from the shallow sector where open water without shore obstacles prevails (not shown). A fit on IWHS (Fig. 8b) gives correlation coefficients of $R = 0.79$ and $R = 0.84$.

The results presented above suggest a strong correlation between the spatially distributed wave heights from which the WPI is calculated, and the point measurement of water level variations at the shore from which the "wave energy" is quantified with the IWHS. However, it is important to note that the low number of points associated with the uneven distribution of values results in a heterogeneous point weight in the linear regression. The latter is therefore significantly affected by isolated values (e.g. by the highest WPI value). A larger data set would be required to strengthen this relation.

### 4.3 Meltwater plume occurrence

Figure 9 shows peaks in meltwater plume occurrence of different amplitudes and widths along the calving front. A peak of meltwater plume occurrence of 95 % spreads over more than 500 m along the deep sector of the calving front. The second highest peak (67 %) is located in the shallow sector and is about 300 m wide. Six smaller peaks were detected in both sectors with plume occurrence of up to 38 %. On average, meltwater plumes occurred 18 % of the time in the shallow sector, and 38 % of the time in the deep sector. The likely explanation for this more than two times higher occurrence is that the bedrock

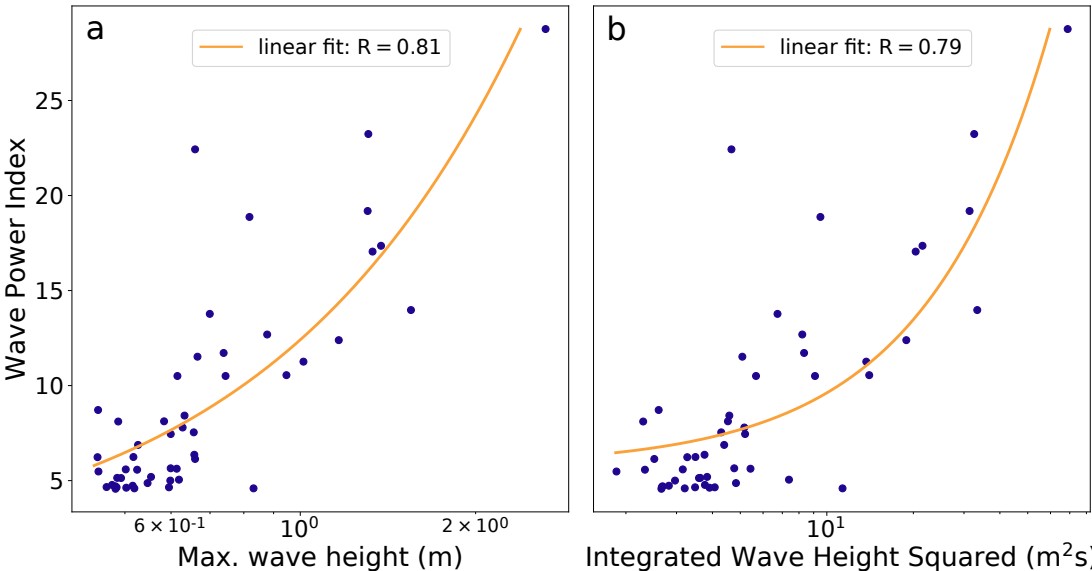

**Figure 8.** (a) Relation between wave power index (WPI) and maximum wave height and (b) Integrated Wave Height Squared (IWHS) for the 50 biggest waves recorded by the pressure sensor, on linear-log scales. The IWHS (measure of wave energy) is computed within a $\pm 3$-minute interval around the maximum wave height. The orange lines correspond to linear fits using linear least squares applied to each data set and are associated to their respective $R$ correlation coefficients.

topography guides subglacial channels preferentially into the deep sector, assuming that the bathymetry extends inland of the calving front. The relation between WPI and meltwater plume occurrence is discussed in section 5.3.2.

## 5   Discussion

We presented a novel method (TeRACWA) for the detection and the quantitative assessment of calving activity by analysis of calving waves recorded with a TRI. This method is complementary to other methods such as calving volume estimates by subtraction of Digital Elevation Models (DEMs) derived from drone imagery or from TRI interferometry (Walter et al., 2020; Cassotto et al., 2019; Xie et al., 2019), from time-lapse photography (e.g., Minowa et al., 2018) or from seismic recordings (e.g., Amundson et al., 2012; Walter et al., 2012; Winberry et al., 2020).

Each of these methods detects different aspects of the calving process. High-rate TRI interferometry provides detailed calving volumes and the locations of sources above the water line, but submarine calving events currently escape detection largely because of the loss of signal coherence over the ocean surface. Analysis of high-rate time-lapse photography provides dense coverage of events, but is difficult to quantify, and can be challenging to automate. Calving waves at remote shores provide estimates of the calving impact on the ocean, but are difficult to interpret in terms of ice volume and source location

due to various calving styles and wave propagation phenomena. Passive seismology captures mainly large events with distinct

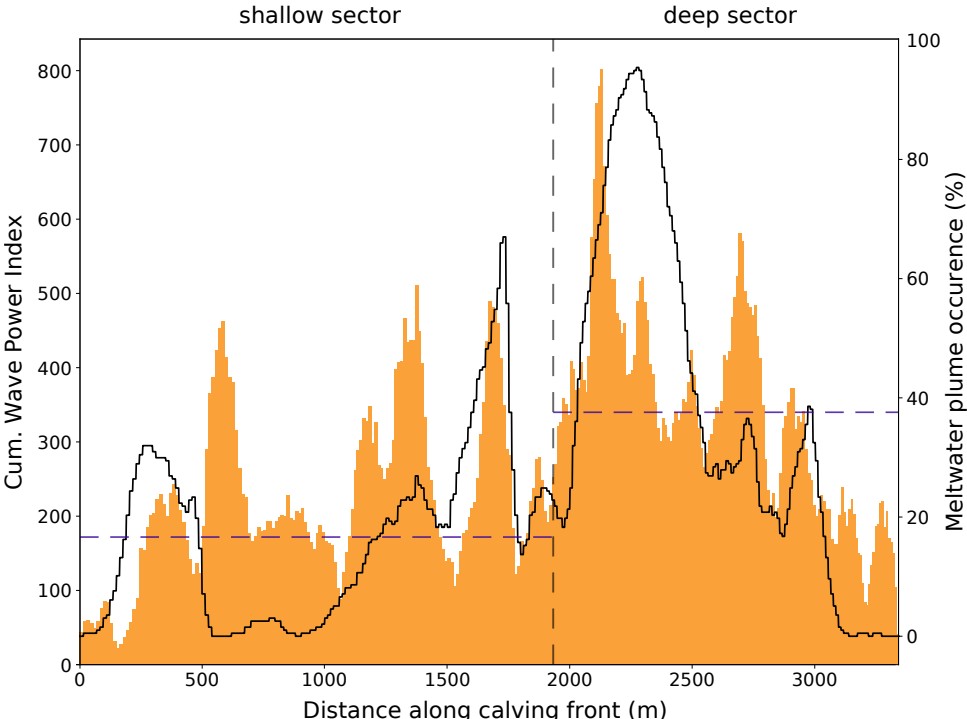

**Figure 9.** Temporally cumulated wave power index (orange bars; identical to Fig. 5d) and fraction of meltwater plume occurrence (black line) along the calving front. Shallow and deep sectors are separated by the the vertical gray dashed line. Horizontal purple dashed lines represent the average meltwater plume occurrence for each sector.

fracturing and ice-rotational processes (e.g. Walter et al., 2020; Sergeant et al., 2019), whereas cliff failure and submarine calving produce seismicity mainly through wave action.

Our novel TeRACWA method detects calving waves from all calving styles and reliably provides source location and timing. The method detects secondary effects of the calving process, and therefore yields information not available from other methods. At present, however, the method cannot discern between different calving styles. A combination of several methods would be a promising avenue to a clearer understanding of the whole calving process including fracture, ice failure, ocean impact and wave propagation.

## 5.1 Validation of the TeRACWA method

In this section, we discuss challenges associated with the method validation, and specifically the comparison of TeRACWA results with the pressure sensor measurements and with another TRI-based calving event detection method. Finally, a possible validation setup is proposed.

A comparison of the WPI determined by TeRACWA with wave amplitudes derived from pressure measurements was presented in Section 4.2. While the pressure sensor monitors the water surface on the opposite shore of the fjord at $3.5\,\mathrm{km}$ distance from the region of interest, the TRI provides observations at the front of the glacier. Since waves undergo various modifications linked to the fjord bathymetry and shore geometry, and are damped by the variable floating ice debris, their properties continuously evolve from the generation point to the pressure sensor. The high correlation between the two signals (Fig. 8) therefore cannot be used as a rigorous validation of the wave power quantification, but still is reassuring.

The two TRI-based calving detection methods TeRACWA and the surface elevation change extraction method (SECEM) (Walter et al., 2020) thereafter compared are based on the same TRI data acquisitions. SECEM detects calving events as changes in ice surface elevation from TRI-derived digital elevation models, and provides exact timing, location and volume of individual calving events. Ideally, the calving events detected with both methods would agree. Several attempts at a direct comparison showed important differences which are due to the different aspects of the calving process captured by each method. While this difference precludes the validation of one method with the other, it illustrates the complementary character of both methods which are derived from the same data set. The requirements for similar calving detections, and the main reasons for the differences, are discussed below.

The first requirement is that both methods capture the same calving events. This is certainly fulfilled for big subaerial changes in ice volume, as each large chunk of ice falling into the fjord creates a wave detected by TeRACWA with a height linked at the first order to the ice volume. However, there is no direct per-event proportionality between falling ice volume and the height of the resulting wave, which depends on the elevation above water of the detached ice mass and details like impact angle, fragment size and water depth. The local terminus geometry such as the front slope and the presence or absence of bedrock above the water line before any contact with the ocean surface are also important parameters affecting the wave properties. Furthermore, not all detected waves stem from changes in subaerial ice volume. Submarine calving events can only be detected by their wave action through TeRACWA without a counterpart in the SECEM results.

The second requirement is that events of all sizes are detected by both methods. Both methods feature a lower detection limit to prevent false detection of noise. These thresholds, 5 m in height for SECEM DEM differentiation and 4.5 in WPI for TeRACWA, are based on different quantities and are therefore not suppressing the same events.

Finally, it is important to note that SECEM involves a temporal stacking of 10 minutes in order to reduce noise from atmospheric disturbances. Events within each stacked period are therefore merged together requiring resampling of the TeRACWA results for a meaningful comparison.

Despite these method differences, we find similarities in the results from TeRACWA and SECEM (Walter et al., 2021). At a 10-minute temporal resolution, both methods detect the same large calving events in space and time. In contrast, we found no clear similarities for small events, again likely due to the above-mentioned methodological differences and due to the method limitations discussed in the next section. SECEM detected larger ice volume changes in the deep sector than the shallow sector (+26 %) which can be connected to the higher temporally cumulated WPI determined by TeRACWA in the deep sector since the two variables exhibit a clear correlation when averaged over several days. However, a higher number of calving events was detected by SECEM in the shallow sector (+65 %) while TeRACWA pointed out the deep sector as the most active (+10 %).

We suggest part of this deviation to be linked to the TeRACWA capability to detect submarine events that remain undetected with SECEM as well as to the method differences presented above. Both TeRACWA and SECEM highlighted an increase in calving activity during the field season.

One possible approach to robustly validate the wave detection algorithm in space and time as well as and the inferred wave power would be the installation of a pressure sensor or a GNSS Wave Glider (Penna et al., 2018) attached to an anchored buoy at a shorter distance to the glacier front. Calving waves could be monitored close to their source location with lower attenuation and interference. With additional high-rate time-lapse cameras on both banks for a continuous coverage of the glacier front, wave heights, timings and locations could be accurately measured in-situ.

## 5.2 Detection challenges and method improvements

Directly detecting calving waves is challenging for a number of reasons. Calving waves are a transient phenomenon and leave no trace after their dissipation, thus preventing temporal stacking of radar acquisitions for noise reduction. A small sampling interval and low atmospheric disturbances are thus mandatory for sufficiently high-quality data. Due to wave speeds exceeding $30\,\mathrm{ms}^{-1}$ (Lüthi and Vieli, 2016) the one-minute sampling interval of our data set is barely small enough to observe the same wave on several acquisitions, as it travels some $1800\,\mathrm{m}$ between acquisitions. While we are confident that all waves were captured, more frequent sampling would be preferable to detect the waves at an early stage to obtain a more precise localization of the source area and detailed wave characteristics. Attempts were carried out to distinguish between different calving styles by not only analysing the maximum amplitude of the Fourier Transform but also the associated frequency. However, no clear patterns could be identified, which might be linked to an insufficient sampling rate.

The main limitation of the proposed method is linked to the heterogeneous properties of the proglacial marine environment, both spatially and temporally. The radar signal scatter intensity strongly depends on position, size and shape of natural reflectors. Icebergs and small pieces of floating ice debris are continually shifting, driven by wind, tides and subglacial melt water plumes. Consequently, the recorded scattering intensity of a calving wave propagating along a rough ice-covered fjord surface will be significantly higher than that of an ice-free and smooth water surface. In the study case presented here, cold conditions with a high ice cover during the acquisition period alleviated this limitation, which nevertheless has to be carefully taken in account for warmer years like 2019. Signal normalization (TeRACWA step 4) compensates for the effects of variations in radar intensity caused by varying ice cover of the fjord. In addition, differencing of the raw signal of consecutive acquisitions (TeRACWA step 2) significantly reduces the imprint of stable or slow-moving ice-mélange and icebergs, albeit not from highly dynamic areas like the meltwater plume. While these problems reduce the accuracy of the derived wave intensity (WPI) they do not affect the wave detection itself in the case of high wave amplitudes. For low wave amplitudes resulting in WPI values close to the WPI threshold, the ice cover can be determinant in the classification of a signal as wave or background noise. Nevertheless, the WPI threshold is set automatically based on the study of the data set distribution. We therefore suggest the automatic adjustment of the threshold depending on the data set to reduce the influence of this limitation. Applications of TeRACWA for different ice cover conditions in future work will however be needed to further assess this influence. An accurate temporal and spatial tracking of the ice cover motion could improve the normalization of the radar intensity and the WPI determination.

Unfortunately, none of the many tested methods provided good results and high efficiency, such that we decided to use the simple corrections of algorithm steps 2 and 4. For our application of TeRACWA in summer 2018 the strongly ice covered Eqip Sermia bay largely alleviated these limitations.

A further potential error is due to the portion of the pixel mask upstream of the average calving front and extending onto the glacier. There, crevasses scatter the radar signal and contribute to the WPI. The differencing of consecutive acquisitions (TeRACWA step 2) as well as the quantification of the background signal (TeRACWA step 3) reduce the recorded signal from these uninteresting scatterers. Nevertheless, changes of the glacier geometry, such as from ice motion and calving, affects the signal used for wave detection. Ideally, the ocean/glacier interface could be detected at high temporal and spatial resolution and therefore alleviate the need of the buffer zone on the ice. For this purpose, an automatic calving front detection algorithm was

developed based on the analysis of abrupt signal strength changes from fjord water to ice. However, a temporal stacking over several hours was needed to retrieve a clear signal without the influence of icebergs and ice-covered areas. Using a delineation of the glacier terminus at such low time resolution would result in an abrupt and unreal evolution of the ROI when switching from a stacked period to the next, strongly affecting the consistency of TeRACWA results over time. Consequently, the simple and static extraction of the glacier terminus (TeRACWA step 0) was used.

The method accuracy is questionable in the case of multiple large calving events in rapid succession in a restricted area (within several minutes and few hundred of meters), an uncommon but possible situation. Such event sequences induce wave superposition and create complex wave patterns that are difficult to disentangle. For such cases, our method (based on a one-dimensional Fourier transform) often struggles to distinguish the different waves due to lack of spatial information. Trials with 2D Fourier transforms showed no clear results, likely due to the inconsistency between the dimensions of the radar images at

different spatial resolution, and with azimuth lines acquired sequentially at different times.

    The TRI viewing angle with respect to the calving front is a potential source of uncertainty. As waves propagate in circles, and the Fourier transform amplitude is maximum at their center, results are not sensitive to small variations in calving front orientation. However, in portions of the glacier front with extreme orientations (e.g. approximate azimuth angle of 24° in Fig. 2, oriented towards the top of the radar image) the wave detection may suffer from shading and deflection effects.

Following a careful assessment of the different limitations presented in this section as well as a possible adjustment of the cutoff wavelengths, the proposed method could be applied to other outlet glaciers with various calving styles. This would allow for a better understanding of the factors affecting the results and consequently render the method more robust. Ultimately, the method should become applicable to glaciers with various front geometries, bathymetries, and ocean ice cover.

### 5.3   Glaciological observations

In this section we discuss and interpret the spatial and temporal evolution of the observed calving wave activity. Special emphasis is given to the influence of meltwater plumes on the calving process and calving activity.

### 5.3.1 Evolution of calving wave activity

A long-term increase in spatially cumulated WPI was observed (time correlation of 0.68) during the 2018 field season. However, we found no direct relation with air temperature, humidity, or shortwave radiation, suggesting no direct meteorological influence. Also, no evidence for an influence of tides on calving was found during the 7-day period. These conclusions support earlier findings (Walter et al., 2020) that variations in calving activity at Eqip Sermia have no obvious short-term relationships to external forcings. We interpret this short-term independence from environmental forcings as the signature of fracture processes that are driven by internal state (such as stress state and pre-existing weaknesses) and are active on intrinsic time scales. Short-term variations in environmental parameters like diurnal temperature or semi-diurnal tides may occur at frequencies that are too high to systematically affect the calving process. Alternatively, the amplitude of the these forcings may be too small to trigger calving processes. Since only direct, linear and short-term influences have been assessed here, we cannot rule out more complex, delayed or cumulative relations affecting calving.

A marked spatial variation in calving wave activity was observed along the glacier front. In the deep sector the average temporally cumulated WPI was 34 % higher at a 26 % smaller width as compared to the shallow sector. Normalized by sector width this difference amounts to 49 %, illustrating two distinct calving regimes (Fig. 6). The marked difference in average water depth between the two sectors likely influences the observed calving wave activity. More precise information on the fjord environment, such as accurate water depths at the calving front, would be necessary to pursue this interpretation. Walter et al. (2021) highlighted strong variations in ice velocity during the field campaign along the calving front (mean of $9.4\,\mathrm{md}^{-1}$ with values from 3.5 to $15.5\,\mathrm{md}^{-1}$), with no clear differences between the shallow and deep sectors. They further identified a significant retreat of the terminus over a large section of the deep sector, the location of maximum retreat (at an approximate along-front distance of $2150\,\mathrm{m}$ in Fig. 5a) coinciding with the area of highest calving wave activity determined by TeRACWA. A smaller area of calving front retreat was identified in the shallow sector and can be linked with a local maximum of calving wave activity (at an approximate along-front distance of $1600\,\mathrm{m}$ in Fig. 5a). In these two cases, the ice flow was therefore not high enough to compensate for the frontal ablation and thus, to maintain a stable terminus position. No clear variations in terminus position were observed between along-front distances of 0 and $1200\,\mathrm{m}$, illustrating a balance between the two latter processes.

### 5.3.2 Calving enhancement by meltwater plumes

Subglacial discharge of melt water into the ocean forms rising plumes and increases submarine melting by entraining warmer ocean bottom water to the calving front. Such submarine melting has been identified as an enhancing mechanism for calving and potential glacier destabilization (Bartholomaus et al., 2013; Fried et al., 2015; Luckman et al., 2015). We further combine our data set of calving wave activity with the manual detection of meltwater plumes in order to investigate their influence on calving activity.

Comparing the temporally cumulated WPI with the presence of meltwater plumes yields a clear relationship. Figure 9 shows that calving wave activity was high where meltwater plumes were often observed. However, there are sections of the calving

front with high calving activity but low visible plume activity. Meltwater plumes might be active but not reach the upper water surface, indicating that their visible footprint might not be sufficient to quantify their strength (Jouvet et al., 2018). We further found that the calving activity is on average 1.77 times higher in the presence of a visible meltwater plume (1.35 and 1.84 in the shallow and deep sectors respectively). The increase in calving activity therefore is 34 % higher in the deep sector than in the shallow sector. We found similar ratios using the number of waves instead of the temporally cumulated WPI (average of 1.71, respectively 1.85 and 1.31 for the two sectors). A similar increase of 70 % in the number of calving events was found at Store glacier ($\sim 60\,\text{km}$ north of Eqip Sermia) in July 2017 (Cook et al., 2020). While we find a similar enhancement of temporally cumulated WPI (77 %), only a 3 % increase in calving volumes was observed at Store glacier.

The strong correlation between meltwater plumes and calving activity, especially in the deep sector, can be explained by the occurrence of large submarine plumes and resulting melt undercutting. Along a deep submerged calving front more warm and salty deep water is entrained in meltwater plumes and rises along a larger exposed area of calving front (Rignot et al., 2010), therefore maximizing the influence of meltwater plumes. In contrast, the small water depth of the shallow sector may block the advection of warmer subsurface fjord waters and therefore limit submarine melting.

From these observations we conclude that an important part of the higher calving activity in the deep sector is explained by a combination of a higher occurrence of meltwater plumes and a more efficient heat exchange with warm rising waters. These conclusions could be improved by determining the change rate of the plume footprint area on the fjord surface which gives an estimate of water flux within the melt water plume. To this end an automated detection algorithm is needed with high temporal and spatial resolution. Our attempts to develop such a method based on watershed algorithms were unsatisfactory due to an insufficient detection accuracy. The main limitation was the difficulty to accurately detect the calving front on single radar acquisitions (cf. Section 5.2). While the automatic tracking of growing meltwater plume extent at known locations was successful, the detection of newly formed plume footprints was complicated by low-backscatter features on the glacier such as crevasses.

The above discussions and interpretations are based on unique high resolution observations during a 7-day period. Such a short time period captures only a short-lived snapshot of the very dynamic processes at the calving front during the melt season. Our observations only capture a limited range of environmental conditions and processes, and more processes and changing dynamics might be active throughout the melt season. Likely larger differences in calving regimes can be observed throughout the summer, or between years. Longer and more frequent continuous field observations are therefore crucial to study calving processes in a variety of hydro-meteorological and environmental contexts. Our understanding of the complex calving phenomenon and its various implications hinges on precise high resolution observations with many complementary methods.

## 6  Conclusions

We developed a novel automated method named TeRACWA (Terrestrial Radar Assessment of Calving Wave Activity) for the detection and the quantification of ocean waves generated by glacier calving. Using radar scatter intensity from a terrestrial

radar interferometer (TRI), the algorithm yields timing and source location of calving events, as well as a measure of wave power quantified by a unit-less wave power index (WPI). It offers the new possibility to detect submarine calving events, a

calving style imperceptible with other TRI algorithms. This method was successfully applied to ∼11500 acquisitions with a TRI during 7 days in a fjord mostly covered by floating ice fragments. We found a higher calving wave activity at the glacier front sector ending in deep water than in shallow water, further correlated with a higher occurrence of visible meltwater plumes. We therefore explain the higher calving activity in the deep sector by a combination of more frequent occurrence of meltwater plumes and a general increase of submarine melting in deeper water.

The recognition of calving events from wave patterns is complementary to other calving detection approaches such as source identification from time-lapse photography, volume estimates from DEM differentiation, or analysis of seismic signals. By using complementary information on various aspects of the calving process from different methods, this crucial process can be studied in more detail. In addition to the new possibility to detect submarine calving events, our method can be applied for re-analysis of existing TRI data sets.

The calving process is a complex phenomenon that can only be investigated in depth by combining different complementary observation approaches. In this way, process understanding from the analysis of high resolution in-situ measurements constitutes a major tool to constrain detailed numerical calving models. Ultimately, this will yield a better understanding of calving dynamics, which is crucial in high resolution ice sheet modeling for assessing the future evolution of the major ice sheets.

*Code availability.*   The implementation of the TeRACWA method is presented in a Github repository: https://github.com/AdrienWehrle/TeRACWA

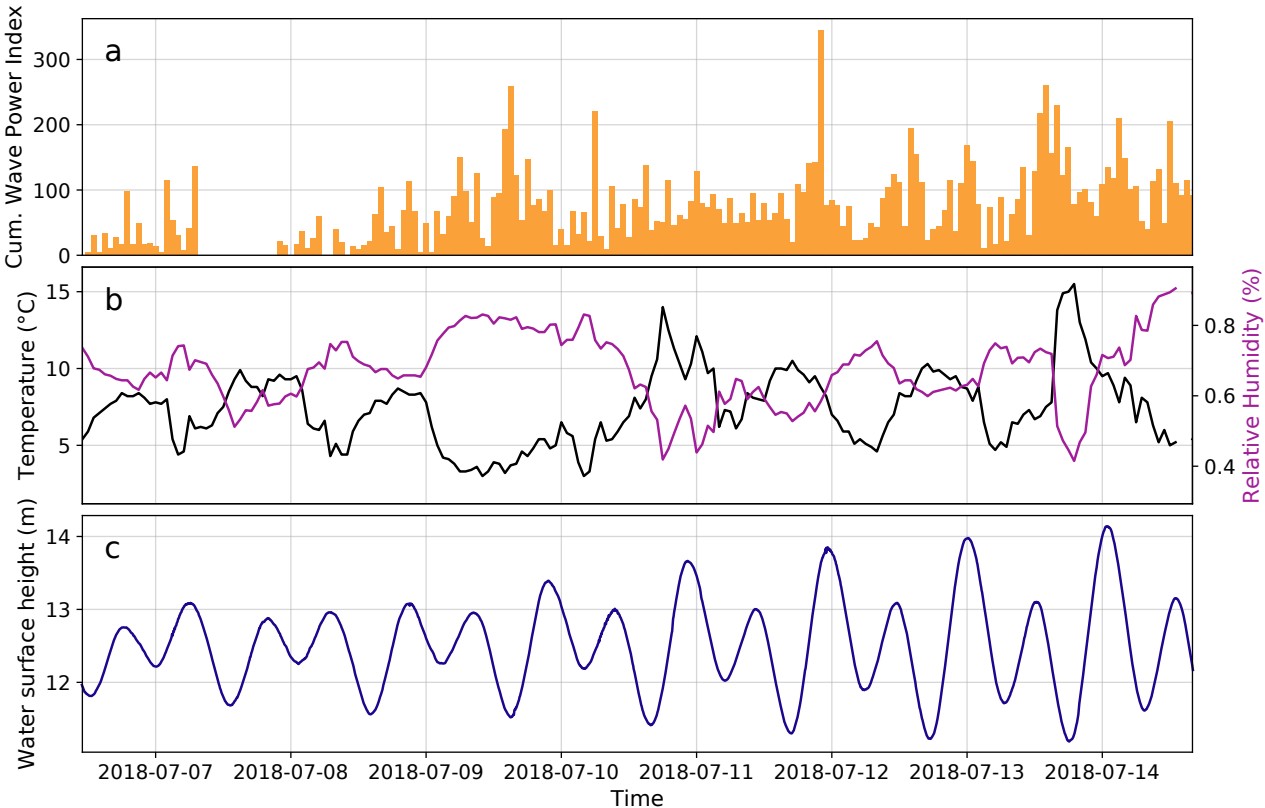

**Figure A1.** (a) Spatially cumulated Wave Power Index determined with TeRACWA here presented as hourly sums, (b) air temperature and relative humidity recorded hourly. (d) Tide heights throughout the acquisition period obtained by applying a low-pass filter to the pressure sensor data with a 0.001 Hz frequency cut-off then resampled to 5 minutes.

**Appendix B**

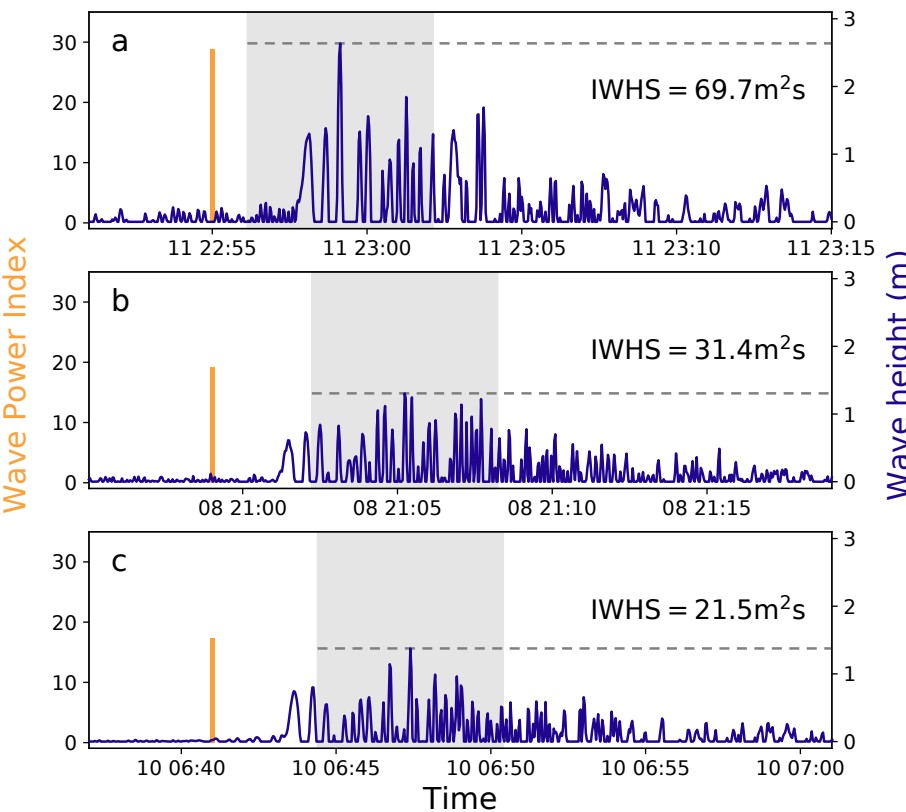

**Figure B1.** Timing of the three highest Wave Power Indexes (orange bars; a to c in WPI descending order) determined by TeRACWA and associated wave heights derived from pressure sensor data (blue curves). Gray dashed lines indicate the maximum wave heights and gray areas, the $\pm 3\,\mathrm{min}$ intervals around maximum wave height used in the IWHS computation.

## Appendix C

**Movie C1.** Animation of consecutive one-minute interval radar images from July 9, 2018 at 13:00:00 UTC to July 9, 2018 at 22:00:00 UTC represented as the logarithm of the signal strength. Clearly visible is the evolution of a meltwater plume footprint in the deep sector, pushing the ice debris coverage away from the calving front therefore creating an open water area. Short-lived wave trains generated by falling ice chunks along the calving front are also discernible as they propagate through the ocean ice cover.

*Author contributions.* AWe drafted the manuscript and performed the data analysis. AWe, ML, AV and AWa performed the field measurements. GJ performed and analyzed the drone flights. All authors contributed to the editing and reviewing of the manuscript. All authors have read and agreed to the published version of the manuscript.

*Competing interests.* The authors declare that they have no conflict of interest.

*Acknowledgements.* We thank Diego Wasser, Eef van Dongen and the UZH Field Excursion students for their help during the field campaign at Eqip Sermia. This work was supported by Swiss National Science Foundation grants 200021_156098 and 200020_197015, and University of Zurich Forschungskredit, grant FK-19-090 to A. Walter.

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
