# Peer review of "Automated detection and analysis of surface calving waves with a terrestrial radar interferometer at the front of Eqip Sermia, Greenland"

_The Cryosphere, 2021_

## Referee Comment (RC2)

**Summary:**

The authors introduce a novel technique to detect tidewater glacier calving events from a terrestrial radar interferometer (TRI). Their algorithm detects surface waves along the fjord surface that occur immediately following iceberg calving events. The authors apply their methods to observations acquired in July 2018 along Eqip Sermia, Greenland. They point out that the new method complements previous calving detection methods because it identifies, both spatially and temporally, the location of calving events along the front, and in particular, is the first to detect submarine calving events. They compare their findings against changes in water surface height derived from a water pressure sensor and find good linear relationship with those observations; thus, providing confidence for the technique.

Overall, the authors have presented a well-written and coherent manuscript with an interesting new algorithm that can be applied to other tidewater glaciers. Cryospheric applications of TRIs continue to evolve, and this manuscript demonstrates a novel use of TRI for calving event detection and source location. Many studies have demonstrated that frontal ablation accounts for nearly half of all ice mass lost in Greenland, yet adequate observations of calving processes remain elusive. Diverse calving mechanisms and the short time scales over which these processes occur inhibit observations. Here, the authors present techniques to assess a specific calving style (frequent and abundant calving of small ice chunks) that are nearly impossible to quantify in the field or through remote sensing techniques. Thus, this technique indeed complements other calving identification methods (photogrammetry, tide pressure gauges, seismometers) by capturing these small-scale events that accumulated over time, can represent significant portions of discharged ice. This manuscript will make a fine contribution to *The Cryosphere.* There are several issues that should first be considered. Most concerns are minor and could easily be addressed by providing additional details on the types of calving events observed. A general characterization of the calving event styles (submarine, subaerial, serac collapses, rotating slabs, etc) and approximate quantities of each type observed should be included, if possible, to help convey the utility of this new technique. My comments in the General and Line-by-Line sections below are provided to elucidate where these areas could be addressed in the manuscript.

**General Comments:**

1) A general characterization of the calving event sizes and styles should be included. The authors detected 2418 calving events over 7.5 days, equivalent to 1 calving event approximately every 4.5 minutes. This seemingly high number of calving events could have affected the geometry of the calving front, yet no discussion on the impact to the calving front is provided. *Did these events cause a change in terminus front position?* No estimate of glacier speed is provided; however, an earlier paper by several co-authors showed flow rates ranged from 1 to 6 m d$^{-1}$ in Aug 2016. Assuming similar flow rates occurred in Jul 2018, the >2000 calving events observed would seemingly induce a significant retreat of the calving front, consistent with the "rapid retreat and flow acceleration" mentioned on L49. To be clear, a thorough characterization of all 2418 events is probably beyond the scope of this study (to introduce a novel calving event detection technique from fjord surface waves); however, additional context about the general size and style of calving events observed is warranted. This would help contextualize the types of calving events detected by this method. A simple characterization would further emphasize the advantage of this new technique to quantify small calving events (see #6 below). In addition, the authors might consider adding the minimum and maximum calving front positions to Fig 2 to demonstrate any change in terminus position, or lack thereof.

2) The authors repeatedly mention the utility of their technique for detecting submarine events; however, it remains unclear how many events were actually submarine in origin or how the surface wave signature from a submarine event may differ from subaerial events and rotating slabs. Their algorithm is based on

the power spectra from 30 wave signals (~1% of the detected calving events). *Were these events subaerial or submarine events? Were they of similar event size? How would this algorithm change for larger calving events observed at other tidewater glaciers?* For example, the authors invoke an 800 m upper wavelength cutoff to suppress reflections from the shore. Assuming that bigger calving events produce bigger and longer surface waves, *how could future users differentiate between such events and reflections from the shore? Is this technique limited to tidewater glaciers that only produce small calving events?*

3)  The authors use an integration interval of 6 minutes (+/- 3 minutes from maximum wave height) to calculate the Integrated Wave Height Squared (IWHS). This time interval was specifically chosen to "capture most of the wave energy without overlap with following waves" (L142). If calving events occurred, on average, every 4.5 minutes, *how do the authors prevent aliasing from events within short time intervals (e.g. < 6 minutes apart)?* Could the authors please clarify this?

4)  The use of TRI for meltwater plume detection is itself novel and intriguing and could be quite valuable to the ice-ocean and fjord circulation communities. However, important details are missing, which should be included. On L148-150, the authors define plume detection as the absence of ice debris coverage (mélange) in azimuth lines along the calving front. *How do the authors discern between plume activity and mélange dispersal due to calving and/or wind events?* The images provided (Fig 1, 2, 5) show the ice mélange at Eqip Sermia appears as a veneer of small iceberg bits floating along the surface that does not occupy the entire fjord (Fig 1). Similarly, both Fig 1 and 5 show alternating areas of sediment rich and clear water. Given the varying surface conditions and lack of an extensive ice mélange that occupied the entire fjord width, *how do the authors differentiate between open water that appears naturally (i.e. in the absence of plumes) from plume driven polynyas?* Finally, including a few of the 195 hourly MLI image stacks could be useful for conveying plume detection with the TRI.

5)  Given that several comparisons between the shallow and deep sectors of the bay are made in the manuscript (e.g. Figs 1, 5, 6 and 9 and throughout the text in the results and discussion), the authors might consider quantifying or providing some constraints on the depths in these regions. Walter et al, (2020) demonstrated that sections of the shallow region are actually above sea level. Calved ice falling onto shorelines are likely to create very different surface waves than ice falling into deep water. *Do the authors, in fact, see differences in the surface waves along the very shallow shorelines and those generated in deeper water?* Regardless, some estimate of bed elevation would be useful.

6)  The proposed technique presents an obvious advantage over other methods: to quantify calving losses produced from frequent small calving events that are not captured by other methods, including earlier TRI techniques that used similar radar return images to quantify geometric changes along calving fronts. Specifically, this new technique capitalizes on the radar's ability to detect fjord surface waves generated from calving events that produce icebergs that are smaller than the resolution of the radar (e.g. icebergs <32 m in a single dimension at a 4.5 km slant range distance) at temporal sampling rates that are orders of magnitude higher than can be obtained by satellite observations; thus, the technique measures ice mass loss that would otherwise go undetected. As shown in earlier work by some of the co-authors, several thousand of these small events integrated over a 1-week period could produce ice discharge values comparable with fewer but larger events at other glaciers. The major advantage of capturing these small-scale calving events to produce a comprehensive calving record is alluded to in the manuscript, but never stated explicitly. The authors should consider highlighting this benefit to readers.

**Line-by-line comments:**

L14 & L16: Ice Sheet should be capitalized when used as a proper name (i.e. preceded by Greenland).

L41: Kane et al,(2020) used surface waves to detect calving. The current manuscript quantifies this technique in new and exciting ways. Nonetheless, the Kane reference should probably be included here.

L50: "associated with high calving…".

L62: The GPRI's effective resolution is 7 m at 1 km slant range, not 4.5 km.  Please correct.

L67: This statement could be written more concisely. Recommend changing to "..water surface height to retrieve the amplitude and timing…"

L71: Only the signal strength (amplitude) is used in this study, the phase is not. Consider removing the mention of phase to avoid confusion or specify that only the amplitude images are used.

L72: raw radar acquisitions "were" stored as…

L83: recommend change "of" to "from"

L91: Do you multi-look the MLI images? If so, please indicate how many pixels in range and azimuth are multi-looked.

L94: "differenced"

L96: Here and elsewhere, wavelengths should be one word, not two.  (see also L99, L104)

L97: Only the electromagnetic phase measurements are affected by atmospheric noise.  The MLI images (signal amplitude) are not. Since this study only uses MLI images, there should be no need to filter atmospheric noise. Please clarify or remove this statement.

 L147-154: As mentioned above, this section would benefit from additional details to bolster the use of TRI for plume detection.

L170: How are wave widths defined? At the time of impact? A maximum width after some length of time? Please clarify.

L175-180: Do you have a figure to support this correlation? It is hard to visualize this relationship.

L194-195: Why are these statistical relationships not shown in Fig 8? Including the linear LSQ fits would be helpful.

L195: Can you explain what you mean by "where open water without obstacles prevails"? What obstacles are you referring to? Bedrock? Sediment shoal?

L238: Recommend removing "The". As written, it appears as though there are only two TRI-based calving detection methods. Note that the use of TRI MLI images has been used to detect calving before.  See, for example, Lüthi et al, (2016), Cassotto et al, (2019), and Kane et al, (2020).

L273: Why are low atmospheric disturbances mandatory if the only TRI product used in this technique is the amplitude of the returned radar wave (i.e. the MLI)?

L279-291: It would seem that in the Ku-band, a significant fraction of the fjord nearest the terminus would need to be sufficiently covered by iceberg bits, an ice mélange veneer, to obtain sufficient radar returns from the fjord water surface. This is alluded to in the paragraph, but not stated explicitly. This seems like an important limitation that should be stated clearly, especially for use in future studies.

L311: Do you mean Figure 2?

L313-315: Perhaps this would be a good paragraph to mention the minimum ice mélange conditions for the detection of surface waves.

L360: Do you mean "Our attempts to …"?

Figure 4: What is the difference between the dashed and solid lines?

Figure 5: "20 minute stacks" – minutes should not be plural in the figure caption. Also, could the authors make the colored lines in panel b (WPI values) thicker to improve readability? Or perhaps modify the color scale to emphasize higher values of WPI? It is difficult to differentiate the higher WPI values (reds, yellows?) from the low values (blue and purples) that appear to dominate the figure.

Figure 6: The value of this figure remains a little unclear. Doesn't this figure show generally the same result as Figure 5? Furthermore, if I understand the figure correctly, it demonstrates that the magnitude of waves is higher in deeper water than in shallow water, but the wave width is smaller. Is this a consequence of ice falling into very shallow water or perhaps partially on land (shoreline)? Walter et al (2020) showed portions of the shallow region are actually grounded above sea level. This could account for differences in calving induced wave activity. There is no discussion whether ice in the shallow section fell partially into the water. Please address this and whatever impacts it may have on fjord surface waves.

Figure 7: Recommend increasing font size of the ticks along both axes.

**Citations:**

Cassotto, R., Fahnestock, M., Amundson, J. M., Truffer, M., Boettcher, M. S., la Peña, de, S. and Howat, I.: Non-linear glacier response to calving events, Jakobshavn Isbræ, Greenland, Journal of Glaciology, 65(249), 39–54, doi:https://doi.org/10.1017/jog.2018.90, 2019.

Kane, E., Rignot, E., Mouginot, J., Millan, R., Li, X., Scheuchl, B. and Fahnestock, M.: Impact of Calving Dynamics on Kangilernata Sermia, Greenland, Geophysical Research Letters, 47(20), 1–11, doi:10.1029/2020GL088524, 2020.

Luthi, M. P. and Vieli, A.: Multi-method observation and analysis of a tsunami caused by glacier calving, The Cryosphere, 10(3), 995–1002, doi:10.5194/tc-10-995-2016, 2016.

---

## Author Comment (AC1)

**Response to reviewers: Automated detection and analysis of surface calving waves with a terrestrial radar interferometer at the front of Eqip Sermia, Greenland**

Adrien Wehrlé, Martin P. Lüthi, Andrea Walter, Guillaume Jouvet, and Andreas Vieli

**September 2021**

We are grateful for the thorough and precise comments made by the two referees, Surui Xie (Referee 1) and Ryan Cassotto (Referee 2), that greatly helped us improving our manuscript. In the following, we address general and detailed comments point by point starting with Referee 1 and proceeding with Referee 2. Answers are written in blue and associated modifications are written in red in the revised manuscript and described here in the same color.

**1 Referee 1**

**1.1 General comments**

According to the authors, the deep-water sector of Eqip Sermia calves more frequent, has a larger average calving size, and has better thermal exchange with the ocean. To my understanding, all these could make the deep sector lose ice faster than the shallow sector. However, the glacier does not show a significant contrast in terminal positions between the shallow and deep sectors (from Figures 2a and 5a and Walter et al., 2020). Why is that? Walter et al. (2020) analyzed a dataset acquired in 2016 while we based our study on the 2018 field campaign, two datasets showing different calving dynamics. In a recent publication, Walter et al. (2021) compared data sets from different years including 2018. Linked to a similar comment from Referee 2, we therefore added more information about the relation between calving activity and front position variations as well as ice flow estimates based on results from Walter et al. (2021) at L367-376 and added the front outlines at the beginning and end of the campaign in Figure 5a.

Section 5.1 listed some differences between the TeRACWA and the SECEM methods. "Despite these method differences, we find similar calving characteristics with TeRACWA as were reported with SECEM (Walter et al., 2020). A higher number of calving events was detected in the deep sector than in the shallow sector." (lines 258-259). And in several other places the authors were trying to echo Walter et al. (2020). However, I think results from the two studies are quite different: more frequent calving events at the shallow sector were reported by Walter et al. (2020), whereas this manuscript shows the opposite. The interannual variability in calving activity at Eqip Sermia is important, such that dynamics from year to year can be very different. Walter et al. (2020) studied the calving dynamics during the 2016 field season while here we used data acquired in 2018, therefore they cannot be directly compared. Nevertheless, in a recent publication, Walter et al. (2021) analysed the results for field seasons from 2014 to 2019 and found higher calving volumes in the deep sector in 2018 (+30%) but also a higher number of events in the shallow sector. The latter result was initially not correctly reported in the manuscript, we therefore now integrated it at L282-288. As presented in the manuscript, we suggest the deviation to be linked to the method differences as well as to the TeRACWA capability to detect submarine events that remain undetected with SECEM.

Was the source really located? The range of calving waves along the radar azimuth may be determined. Need to clarify the source location. The fifth step of TeRACWA gives access to the wave width along the

azimuth dimension but does not assess the range span. A detection in range would have required to split the azimuth lines with a given kernel size on which the Fourier Transform would have been applied. Because the choice of the kernel size was dependent on each wave characteristics and source of additional error, we decided not to apply the detection in range. To clarify this point we changed "the location and width of the wave in space" by "in the azimuth dimension (along front)" (L121) and added "The location therefore consisted in the azimuth lines over which a given wave was detected without further information about the range span." in the next sentence. We also dropped "Only the end products were resampled to a geodetic coordinate system (UTM22N)." (L82) which was misleading as we actually always work with along-front positions in this study.

Rotation or break up of icebergs in the fjord may cause similar waves. Eqip Sermia calves relatively small icebergs. We suggest iceberg break-up and rotation is not detected within the restricted frequency range because the resulting waves are too low frequency and exhibit small amplitudes. We added this clarification at L98.

Some calving events can last several minutes to tens of minutes, they may be counted as multiple calving events by the method. This is one of the motivations behind the sixth step of TeRACWA. The initial goal was to prevent the detection of multiple peaks along the front for a single wave, associated with noise. Because we use a 2D peak detection (in azimuth and time), this also applies to the correction for multiple peaks in time. We precised this point at L133-135.

**1.2** Detailed comments**

Title: Since the manuscript discussed both surface and submarine calving activities, would it be more appropriate to delete "surface" or change to "surface waves generated by calving" or "calving waves on the surface"? We wrote "surface calving waves" thinking of "surface" as associated with "waves" and not "calving". To maintain a reasonable title length, we decided to keep the current one.

Line 32:  $50m \rightarrow 50$  m, and elsewhere, eg., in line 60:  $17.4mm \rightarrow 17.4$  mm, line 179,  $4am \rightarrow 4$  am.... We added spaces between values and units wherever needed.

Line 80: Figure 4 was quoted before Figure 3 in the text. If the journal requires figures to appear in the same order as they are quoted in the text, then the figures need to be rearranged. We swapped Figures 3 and 4.

Line 112, Step 5: What are the typical "WPIs" or peak prominences with false detection when no calving waves exist in the fjord? The typical WPI without any calving wave is on average 1.9. We added this result at L129.

Line 126: Or only one calving wave can be detected if there are multiple calving events which occurred at about the same time. It depends if the peaks are distinct. If they are not, only one wave will indeed be detected. We discuss this point at L334-339.

Line 137: What are the reasons that simpler measures are less suitable? On intuition one may think that the maximum wave height should also be a good measure. And I don't see significant difference in the relationships shown in Figure 8. The maximum wave height does not capture the temporal spread of the wave but only the peak amplitude. The Integrated Wave Height Squared provides more information about the properties of the wave. We agree with the second point of the reviewer, but we still found important to show the results for both measures. We clarified the main difference between maximum wave height and IWHS at L143-144.

Line 169: ... both the average WPI ..., add "average". We included this suggestion.

Lines 217-218: If submarine calving events can be detected by the TeRACWA based on intensity images, TRI phase interferometry should be able to detect some of them (in favorable conditions) since they cause elevation and velocity perturbations in the fjord. The challenges are that TRI interferometry may suffer from temporal decorrelation for velocity estimates and may not be precise enough to detect elevation changes caused by some submarine calving events. Very interesting point. Part of our decision to work on signal amplitude was motivated by the challenge pointed here by the reviewer. We included this point at L236.

Line 219: I think it is subjective to say that time-lapse photography cannot be easily automated. We modified for "can be challenging to automate".

Line 263: Or due to noise. The noise limitation was addressed by the comparison of the different threshold of the two methods few sentences above, therefore included in "the above-mentioned methodological differences" (L281).

Line 311: Fig. 2 instead of Fig. 4? We modified for Figure 2.

Line 388: Some people may disagree. We added "investigated in depth".

Figure 3d: Add a line to show the high-cutoff wavelength? We added a line for the high-cutoff wavelength.

Figure A1: Add one more panel to show the de-tided water surface heights? Readers may be curious to see their correlation with the cumulated WPI. The focus of this figure was more on the evolution of the variables over several days with hourly sums than on the high-frequency changes. For readability and clarity, we decided to only keep the three current panels.

**2 Referee 2**

**2.1 General comments**

A general characterization of the calving event sizes and styles should be included. The authors detected 2418 calving events over 7.5 days, equivalent to 1 calving event approximately every 4.5 minutes. This seemingly high number of calving events could have affected the geometry of the calving front, yet no discussion on the impact to the calving front is provided. Did these events cause a change in terminus front position? No estimate of glacier speed is provided; however, an earlier paper by several co-authors showed flow rates ranged from 1 to 6 m d-1 in Aug 2016. Assuming similar flow rates occurred in Jul 2018, the > 2000 calving events observed would seemingly induce a significant retreat of the calving front, consistent with the "rapid retreat and flow acceleration" mentioned on L49. To be clear, a thorough characterization of all 2418 events is probably beyond the scope of this study (to introduce a novel calving event detection technique from fjord surface waves); however, additional context about the general size and style of calving events observed is warranted. This would help contextualize the types of calving events detected by this method. A simple characterization would further emphasize the advantage of this new technique to quantify small calving events (see #6 below). In addition, the authors might consider adding the minimum and maximum calving front positions to Fig 2 to demonstrate any change in terminus position, or lack thereof. Further work has been carried out to try to distinguish between different calving styles with a focus on submarine calving events by analyzing not only the amplitude but also the frequency of the Fourier Transform computed with TeRACWA. However, no clear patterns (even through manual analysis) could be highlighted. We suggest that this is linked to the 2-minute sampling rate which is not high enough to monitor short-lived events like calving waves in details. We included this point in more details at L301-303. We agree with the reviewer on the need to discuss the interplay between ice flow and calving activity in shaping the calving front (a point also raised by Referee 1). We therefore addressed this point at L367-376 by introducing flow estimates recently published in

**Walter et al. (2021) and by discussing variations in terminus positions between the beginning and the end of the field season, positions that we further added on Figure 5a.**

The authors repeatedly mention the utility of their technique for detecting submarine events; however, it remains unclear how many events were actually submarine in origin or how the surface wave signature from a submarine event may differ from subaerial events and rotating slabs. Their algorithm is based on the power spectra from 30 wave signals ( 1% of the detected calving events). Were these events subaerial or submarine events? Were they of similar event size? How would this algorithm change for larger calving events observed at other tidewater glaciers? For example, the authors invoke an 800 m upper wavelength cutoff to suppress reflections from the shore. Assuming that bigger calving events produce bigger and longer surface waves, how could future users differentiate between such events and reflections from the shore? Is this technique limited to tidewater glaciers that only produce small calving events? TeRACWA indeed also detects submarine events but at the moment, cannot distinguish them from the other calving styles which is mainly due to the sampling rate as described in the answer to the previous comment. Future work could be focused on the distinction between calving styles by e.g. employing a thorough manual analysis of acquisitions from timelapse cameras and improve TeRACWA based on a deeper understanding of submarine events. The wave signals randomly chosen to set the frequency cutoffs cover a large range of amplitudes (from WPIs close to the WPI threshold to WPIs close to the highest WPI of the data set). In combination with timelapse images, we identified 5 of the 17 wave signals to be associated with a submarine calving event (the other 13 waves occurred when the camera was not running). We added this information at L102-104. The sample dataset used here, although indeed small compared to the entire dataset, pictures the main calving styles of Eqip Sermia for events of different amplitudes. We are confident that TeRACWA can be used for other outlet glaciers (point further developed at L344-345) since it is capable of detecting various types of calving events from low to high magnitudes. The cut-offs might have to be adjusted to better target the calving dynamics of a particular glacier (different than Eqip Sermia, e.g. featuring larger events). This could be done in a similar way as in this study, by supervising the cutoff determination with the analysis of a randomly generated sample dataset.

The authors use an integration interval of 6 minutes (+/-3 minutes from maximum wave height) to calculate the Integrated Wave Height Squared (IWHS). This time interval was specifically chosen to "capture most of the wave energy without overlap with following waves" (L142). If calving events occurred, on average, every 4.5 minutes, how do the authors prevent aliasing from events within short time intervals (e.g. < 6 minutes apart)? Could the authors please clarify this? Indeed, the integration interval for the computation of the IWHS remains problematic. The latter can be too short and therefore not fully cover a given wave, or too long thus taking neighbouring waves into account. Here we estimated 6 minutes as a compromise between these two limitations. We worked on a dynamic interval but could not obtain satisfactory results. We clarified this point at L149-151.

The use of TRI for meltwater plume detection is itself novel and intriguing and could be quite valuable to the ice-ocean and fjord circulation communities. However, important details are missing, which should be included. On L148-150, the authors define plume detection as the absence of ice debris coverage (mélange) in azimuth lines along the calving front. How do the authors discern between plume activity and mélange dispersal due to calving and/or wind events? The images provided (Fig 1, 2, 5) show the ice mélange at Eqip Sermia appears as a veneer of small iceberg bits floating along the surface that does not occupy the entire fjord (Fig 1). Similarly, both Fig 1 and 5 show alternating areas of sediment rich and clear water. Given the varying surface conditions and lack of an extensive ice mélange that occupied the entire fjord width, how do the authors differentiate between open water that appears naturally (i.e. in the absence of plumes) from plume driven polynyas? Finally, including a few of the 195 hourly MLI image stacks could be useful for conveying plume detection with the TRI. We agree those points should be clarified. Only meltwater plumes surrounded by ice mélange could indeed be detected in the intensity images as the intensity of the radar signal did not allow the distinction between open water (visible in the absence of plumes) and plume-driven polynyas, unlike optical imagery. This limitation remained minor due to the important ice mélange cover in the fjord during the 2018 field campaign. The formation of polynyas within the ice mélange can however be influenced by other processes than the dynamics of meltwater plumes, such as displacements driven by wind or following calving events as pointed by the reviewer. In both cases, the distinctive patterns of plume-driven polynyas allowed the operator to focus only on those features and discard the latter cases. Indeed, wind-driven polynyas are mostly associated with coherent and large scale ice mélange motion in contrast to plume-driven polynyas that are linked to more local and faster ice mélange motion. On the other hand, calving-driven plumes which are short-term events were highly reduced by the hourly averaging and could be directly discarded if still remaining as the timing and location of calving waves are clearly visible on TRI consecutive images. We precised those points at L156-165.

Given that several comparisons between the shallow and deep sectors of the bay are made in the manuscript (e.g. Figs 1, 5, 6 and 9 and throughout the text in the results and discussion), the authors might consider quantifying or providing some constraints on the depths in these regions. Walter et al. (2020) demonstrated that sections of the shallow region are actually above sea level. Calved ice falling onto shorelines are likely to create very different surface waves than ice falling into deep water. Do the authors, in fact, see differences in the surface waves along the very shallow shorelines and those generated in deeper water? Regardless, some estimate of bed elevation would be useful. During our intensive manual and automated inspection of the dataset we did not detect any recurrent differences in wave wavelengths depending on the sector that we highly suggest is linked to the relatively low sampling rate compared to the typical wave lifetime therefore analysing waves at different stages of their propagation. We highlighted a small difference in wave width depending on the sector that might be linked to the differences in bathymetry but the causality can not be confirmed. However, we do think the fjord geometry (L253-254), water depth (L269), and sections with bedrock above sea level, have an influence on wave characteristics. We further developed this point at L269-271 and added estimations of the water depth by sounding extrapolation for the two sectors at L52.

The proposed technique presents an obvious advantage over other methods: to quantify calving losses produced from frequent small calving events that are not captured by other methods, including earlier TRI techniques that used similar radar return images to quantify geometric changes along calving fronts. Specifically, this new technique capitalizes on the radar's ability to detect fjord surface waves generated from calving events that produce icebergs that are smaller than the resolution of the radar (e.g. icebergs <32 m in a single dimension at a 4.5 km slant range distance) at temporal sampling rates that are orders of magnitude higher than can be obtained by satellite observations; thus, the technique measures ice mass loss that would otherwise go undetected. As shown in earlier work by some of the co-authors, several thousand of these small events integrated over a 1-week period could produce ice discharge values comparable with fewer but larger events at other glaciers. The major advantage of capturing these small scale calving events to produce a comprehensive calving record is alluded to in the manuscript, but never stated explicitly. The authors should consider highlighting this benefit to readers. Thank you for this very interesting point that we now highlight at L243.

**2.2 Detailed comments**

L14 & L16: Ice Sheet should be capitalized when used as a proper name (i.e. preceded by Greenland) We included this modification.

L41: Kane et al,(2020) used surface waves to detect calving. The current manuscript quantifies this technique in new and exciting ways. Nonetheless, the Kane reference should probably be included here. We included this reference.

L50: "associated with high calving...". We included this modification.

L62: The GPRI's effective resolution is 7 m at 1 km slant range, not 4.5 km. Please correct. We have a 0.1 degree azimuth resolution (and not 0.4), thus  $\pi * 4500 * 0.1/180 = 7.9$  m at 4.5 km.

L67: This statement could be written more concisely. Recommend changing to "..water surface height to retrieve the amplitude and timing..." We included this suggestion.

L71: Only the signal strength (amplitude) is used in this study, the phase is not. Consider removing the mention of phase to avoid confusion or specify that only the amplitude images are used. In the same paragraph we

specify that only the amplitude is used ("The method described in the following is focused on the analysis of the signal strength." (L73) and we think it is still important to precise to the reader that the TRI is a phase-sensitive radar, even if we only use the signal intensity here.

L72: raw radar acquisitions "were" stored as... We modified this sentence.

L83: recommend change "of" to "from" We included this suggestion

L91: Do you multi-look the MLI images? If so, please indicate how many pixels in range and azimuth are multi-looked. Thank you for stressing this point. Our method uses directly SLCs without any multi-looking, we therefore replaced the incorrect "MLI images" by "intensity images" throughout the manuscript.

L94: "differenced" We modified this sentence.

L96: Here and elsewhere, wavelengths should be one word, not two. (see also L99, L104) We modified this word wherever needed through the manuscript.

L97: Only the electromagnetic phase measurements are affected by atmospheric noise. The MLI images (signal amplitude) are not. Since this study only uses MLI images, there should be no need to filter atmospheric noise. Please clarify or remove this statement. We did identify dynamic variations in the intensity signal of the SLCs acquisition in the region of interest during periods of unstable weather, especially in the case of low clouds. Indeed, the Ku Band is relatively close to e.g. the X band (10 GHz) used for meteorological applications where the intensity of the reflected radar signal is e.g. used to retrieve rainfall intensity through reflectivity/rainfall intensity (Z-R) relations.

L147-154: As mentioned above, this section would benefit from additional details to bolster the use of TRI for plume detection. We added more details following the general comment on plume detection.

L170: How are wave widths defined? At the time of impact? A maximum width after some length of time? Please clarify. In the description of the Step 5 of TeRACWA is specified "The location and width of the wave in the azimuth dimension (along front) was given as the span over the peak full width at half prominence" (L120-122, minor modification after a comment of Referee 1).

L175-180: Do you have a figure to support this correlation? It is hard to visualize this relationship. In Fig 5c (20-minute spatial stacks through time), we think the increase in WPI is visible. We decided not to include an additional figure since this increase is not discussed further in the manuscript.

L194-195: Why are these statistical relationships not shown in Fig 8? Including the linear LSQ fits would be helpful. We included the LSQ fits and associated R values.

L195: Can you explain what you mean by "where open water without obstacles prevails"? What obstacles are you referring to? Bedrock? Sediment shoal? We modified for "where open water without shore obstacles prevails". Indeed, e.g. the easternmost part of the deep sector is not in direct line of sight with the pressure sensor because of the lateral moraine.

L238: Recommend removing "The". As written, it appears as though there are only two TRI-based calving detection methods. Note that the use of TRI MLI images has been used to detect calving before. See, for example, Lüthi et al, (2016), Cassotto et al, (2019), and Kane et al, (2020). Here "the" was referring to the two methods compared here, we therefore precised by adding "thereafter compared" (L258).

L273: Why are low atmospheric disturbances mandatory if the only TRI product used in this technique is the amplitude of the returned radar wave (i.e. the MLI)? This point was addressed in the answer to the comment on the correction of atmospheric disturbances.

L279-291: It would seem that in the Ku-band, a significant fraction of the fjord nearest the terminus would need to be sufficiently covered by iceberg bits, an ice mélange veneer, to obtain sufficient radar returns from the fjord water surface. This is alluded to in the paragraph, but not stated explicitly. This seems like an important limitation that should be stated clearly, especially for use in future studies. As stated at L313-314, the absence of ice mélange reduces the accuracy of the WPI retrieval but the waves are still detected, therefore not preventing the good applicability of the method. However, this is not entirely true for low wave amplitudes resulting in WPI values close the WPI threshold. Indeed, less efficient reflectors will give a lower WPI which could be lower than the WPI threshold in the case of ice free conditions but would have been above with a higher ice over. We suggest the automatic adjustment of the WPI threshold to reduce the influence of this limitation. We included this precision and pointed it out more clearly for future studies at L314-318.

L311: Do you mean Figure 2? Yes indeed, we modified for Figure 2.

L313-315: Perhaps this would be a good paragraph to mention the minimum ice mélange conditions for the detection of surface waves. We addressed this point at L314-318 and stressed further the applicability for future studies at L344-345.

L360: Do you mean "Our attempts to ..."? We modified for this suggestion.

Figure 4: What is the difference between the dashed and solid lines? In the figure caption is specified "Steps linked by dashed and solid arrows are respectively applied to each TRI acquisition and to the resulting data set."

Figure 5: "20 minute stacks" – minutes should not be plural in the figure caption. Also, could the authors make the colored lines in panel b (WPI values) thicker to improve readability? Or perhaps modify the color scale to emphasize higher values of WPI? It is difficult to differentiate the higher WPI values (reds, yellows?) from the low values (blue and purples) that appear to dominate the figure. We modified for "20 minute stacks". We modified the color scale of panel b to better emphasize the sparse high values by still keeping a good rendering of low values. We carried out many tests of data resampling but 20 minutes remained the best trade-off to get a relatively good rendering of the WPI colormap by keeping the small scale spatial patterns of wave occurrence. We also darkened the background to make light colors (mainly yellow) more visible.

Figure 6: The value of this figure remains a little unclear. Doesn't this figure show generally the same result as Figure 5? Furthermore, if I understand the figure correctly, it demonstrates that the magnitude of waves is higher in deeper water than in shallow water, but the wave width is smaller. Is this a consequence of ice falling into very shallow water or perhaps partially on land (shoreline)? Walter et al (2020) showed portions of the shallow region are actually grounded above sea level. This could account for differences in calving induced wave activity. There is no discussion whether ice in the shallow section fell partially into the water. Please address this and whatever impacts it may have on fjord surface waves. This figure is indeed a summary of the analysis of Figure 5 as analysing Figure 5 is, in our opininion, not straight-forward at first sight. The difference in wave width can indeed be a consequence of the contrasting water depths in the two sectors and we precise information about the detachment event and contact with the ocean in order to simulate wave heights and isolate the water depth influence. We do think various parameters at the calving front influence the wave properties and precised them at L269-271.

Figure 7: Recommend increasing font size of the ticks along both axes. We increased the font size of the secondary axis ticks on the x axis. The font size of the axis labels and major ticks being already almost similar, we decided to keep the same font size for the latter.

**3 References**

Walter, A., Lüthi, M. P., and Vieli, A.: Calving Event Size Measurements and Statistics of Eqip Sermia, Greenland, From Terrestrial RadarInterferometry, The Cryosphere, 14, 1051–1066, https://doi.org/10.5194/tc-14-1051-2020, 2020.

Walter, A., Lüthi, M. P., Moreau, L., and Vieli, A.: Drivers of recurring seasonal cycle of glacier calving styles and patterns, Frontiers in Earth Science, 9, 359, 2021.

---

## Author Response (AR2)

**Response to Editor: Automated detection and analysis of surface calving waves with a terrestrial radar interferometer at the front of Eqip Sermia, Greenland**

Adrien Wehrlé, Martin P. Lüthi, Andrea Walter, Guillaume Jouvet, and Andreas Vieli

October 2021

Line 102: should be "We determined that 5 out of 17..." **We modified accordingly.**

Line 157 :areas close to the water surface, and that independently of the ice cover in the fjord (Figs. 1 and 5a) **We clarified this sentence by modifying the second part of the sentence for "whether the fjord is ice covered or not".**

163: " On the averaged, as well as on consecutive images, " I think this needs to be "on the averaged image" **We modified for "On the hourly averaged images".**

186: I didn't understand the error bars on the temporally cumulated WPI in figure 6. Please explain how the error bars were calculated, and how the significance of the differences was calculated. **The bars correspond to the standard deviation for the dataset of each sector. We obtained temporally cumulated WPIs along the calving front, and then averaged them (and computed the standard deviation) per sector. They are therefore variability bars more than error/uncertainty bars, which is why we simply called them "bars". We therefore modified "average values" for "sector-averaged values" and "standard deviations" for "the associated standard deviations for each sector".**
**The significance of the differences is presented at L186-187: "All the differences presented above have been determined to be statistically significant using a t-test yielding p-values below 0.001 and t-statistics from 3.4 to 21.1."**

214: "These results are associated with a better range coverage" – the meaning of range coverage is not clear. A few more words explaining it would help. **The goal was to note that these results are associated with a more homogeneous data coverage compared to the maximum wave height, with more values in the high range. But after discussion, we think this pattern is only very minor, and therefore this sentence is misleading for the reader. In this way, we decided to delete this sentence.**

---

## Author Response (AR3)

**Response to Editor: Automated detection and analysis of surface calving waves with a terrestrial radar interferometer at the front of Eqip Sermia, Greenland**

Adrien Wehrlé, Martin P. Lüthi, Andrea Walter, Guillaume Jouvet, and Andreas Vieli

November 2021

1. Sorry for not giving a more specific critique of what's needed for figure 6. The manuscript needs to make clear the difference between the average wave power index (6b) and the cumulative wave power index (6c). As I read the manuscript, the cumulative wave power index is just the wave power index times the number of observations, but since the relative values are different, this would appear not to be the case. Likewise, it is not stated in the text how the standard deviations for 6c are calculated, (are they spatial standard deviations? What does it mean to present the standard deviation of a set of points on the same scale as a cumulative value?) This needs to be obvious in the text, without the prompting you gave me in your response. **We fully agree and realized the manuscript was lacking a clear explanation of the temporally and spatially cumulated WPI values. The temporally cumulated WPI is not the WPI values times the number of observations, which would indeed give the same distribution than for the temporally averaged WPI. This variable is in fact the sum of the WPI values through time. The temporally cumulated WPI is therefore quantifying a combination of the number of events and their intensity, while the average WPI is normalized by the number of events hence insensitive to the latter. Looking at Figure 6b, one can see that the WPI in on average a bit higher in the deep sector than the shallow sector. This is clearer when looking at the temporally cumulated WPI (Figure 6c) as more waves were detected in the deep sector and with a higher average WPI, both variables contributing to a larger difference in temporally cumulated WPI between the two sectors. We therefore added a description of the variables processed from the catalog of WPI values right after the description of the algorithm (L136-139). In this way, the variable computation is clear to the reader before any result is presented. We also explained the different results obtained with temporally averaged WPI and temporally cumulated WPI in the Results section (L187-191). We further realized the temporally and spatially cumulated WPIs were refered to as "cumulated WPI" few times in the manuscript. We therefore replaced this general term by the full variable name wherever needed. We finally precised that the WPI presented in Figure 6b is temporally averaged (caption of Figure 6), just as the cumulated WPI. We now think those different modifications made the use of sector-averaged values computed from temporally stacked (averaged of cumulated) variables clear and unambiguous.**

2. "All the differences presented above have been determined to be statistically significant using a t-test yielding p-values below 0.001 and t-statistics from 3.4 to 21.1." To use a test such as this, you need to specify what quantities are compared, what null hypothesis is rejected by the test, and what values are assumed to be independent. An editor's null hypothesis is that a t test is being used incorrectly, and this null hypothesis needs to be disproven with well presented evidence. **We specified what quantities are compared, what null hypothesis is rejected by the test, and what values are assumed to be independent at L193-195.**